# DNA methylation meta-analysis reveals cellular alterations in psychosis and markers of treatment-resistant schizophrenia

Eilis Hannon[1], Emma L Dempster[1], Georgina Mansell[1], Joe Burrage[1], Nick Bass[2], Marc M Bohlken[3], Aiden Corvin[4], Charles J Curtis[5,6], David Dempster[5], Marta Di Forti[5,7,8], Timothy G Dinan[9], Gary Donohoe[10], Fiona Gaughran[11,12], Michael Gill[13], Amy Gillespie[11,14], Cerisse Gunasinghe[5], Hilleke E Hulshoff[15], Christina M Hultman[16], Viktoria Johansson[17,18], René S Kahn[19,20], Jaakko Kaprio[21,22], Gunter Kenis[23], Kaarina Kowalec[16,24], James MacCabe[5], Colm McDonald[25], Andrew McQuillin[2,26], Derek W Morris[10], Kieran C Murphy[27], Colette J Mustard[28], Igor Nenadic[29,30], Michael C O'Donovan[31], Diego Quattrone[5,7], Alexander L Richards[31], Bart PF Rutten[32], David St Clair[33], Sebastian Therman[34], Timothea Toulopoulou[35], Jim Van Os[19], John L Waddington[36], Wellcome Trust Case Control Consortium (WTCCC), CRESTAR consortium, Patrick Sullivan[16,37], Evangelos Vassos[5], Gerome Breen[5,6], David Andrew Collier[38], Robin M Murray[39], Leonard S Schalkwyk[40], Jonathan Mill[1]*

[1]University of Exeter Medical School, University of Exeter, Barrack Road, Exeter, United Kingdom; [2]Division of Psychiatry, University College London, London, United Kingdom; [3]Department of Psychiatry, Brain Center Rudolf Magnus, University Medical Center Utrecht, Heidelberglaan, Utrecht, Netherlands; [4]Department of Psychiatry and Neuropsychiatric Genetics Research Group, Trinity Translational Medicine Institute, Trinity College Dublin, St. James Hospital, Dublin, Ireland; [5]Social, Genetic and Developmental Psychiatry Centre, Institute of Psychiatry, Psychology and Neuroscience (IoPPN), King's College London, London, United Kingdom; [6]NIHR BioResource Centre Maudsley, South London and Maudsley NHS Foundation Trust (SLaM), King's College London, London, United Kingdom; [7]South London and Maudsley NHS Mental Health Foundation Trust, London, United Kingdom; [8]National Institute for Health Research (NIHR), Mental Health Biomedical Research Centre, South London and Maudsley NHS Foundation Trust and King's College London, London, United Kingdom; [9]APC Microbiome Ireland, University College Cork, Cork, Ireland; [10]Centre for Neuroimaging and Cognitive Genomics (NICOG), School of Psychology and Discipline of Biochemistry, National University of Ireland Galway, Galway, Ireland; [11]Psychosis Studies, Institute of Psychiatry, Psychology and Neuroscience (IoPPN), King's College London, London, United Kingdom; [12]National Psychosis Service, South London and Maudsley NHS Foundation Trust, London, United Kingdom; [13]Department of Psychiatry and Neuropsychiatric Genetics Research Group, Trinity Translational Medicine Institute, Trinity College Dublin, Dublin, Ireland; [14]Department of Psychiatry, Medical Sciences Division, University of Oxford, Oxford, United Kingdom; [15]Department of Psychiatry, University Medical Center Utrecht, Utrecht, Netherlands; [16]Department of Medical Epidemiology and Biostatistics, Karolinska Institutet, Stockholm,

*For correspondence:
j.mill@exeter.ac.uk

Sweden; [17]Department of Medical Epidemiology and Biostatistics Sweden, Karolinska Institutet, Stockholm, Sweden; [18]Department of Clinical Neuroscience, Center for Psychiatry Research, Karolinska Institutet and Stockholm Health Care Services, Stockholm, Sweden; [19]Department of Psychiatry, Brain Center Rudolf Magnus, University Medical Center Utrecht, Utrecht, Netherlands; [20]Department of Psychiatry, Icahn School of Medicine at Mount Sinai, New York, United States; [21]Institute for Molecular Medicine FIMM, University of Helsinki, Helsinki, Finland; [22]Department of Public Health, University of Helsinki, Helsinki, Finland; [23]Faculty of Health, Medicine and Life Sciences, Maastricht University, Maastricht, Netherlands; [24]College of Pharmacy, University of Manitoba, Winnipeg, Canada; [25]Centre for Neuroimaging and Cognitive Genomics (NICOG), School of Medicine, National University of Ireland Galway, Galway, Ireland; [26]Division of Psychiatry, University College London, London, United Kingdom; [27]Department of Psychiatry, Royal College of Surgeons in Ireland, Dublin, Ireland; [28]Division of Biomedical Sciences, Institute of Health Research and Innovation, University of the Highlands and Islands, Inverness, United Kingdom; [29]Department of Psychiatry and Psychotherapy, Jena University Hospital, Jena, Germany; [30]Department of Psychiatry and Psychotherapy, Philipps University Marburg/ Marburg University Hospital UKGM, Marburg, Germany; [31]MRC Centre for Neuropsychiatric Genetics and Genomics, School of Medicine, Cardiff University, Cardiff, United Kingdom; [32]Department of Psychiatry and Neuropsychology, Faculty of Health, Medicine and Life Sciences, Maastricht University, Maastricht, Netherlands; [33]The Institute of Medical Sciences, Univeristy of Aberdeen, Aberdeen, United Kingdom; [34]Department of Public Health Solutions, Mental Health Unit, National Institute for Health and Welfare, Helsinki, Finland; [35]Department of Psychology and National Magnetic Resonance Research Center (UMRAM), Aysel Sabuncu Brain Research Centre (ASBAM), Bilkent University, Ankara, Turkey; [36]Molecular and Cellular Therapeutics, Royal College of Surgeons in Ireland, Dublin, Ireland; [37]Departments of Genetics and Psychiatry, University of North Carolina at Chapel Hill, Chapel Hill, United States; [38]Neuroscience Genetics, Eli Lilly and Company, Surrey, United Kingdom; [39]Department of Psychosis Studies, Institute of Psychiatry, King's College London, London, United Kingdom; [40]School of Life Sciences, University of Essex, Colchester, United Kingdom

**Abstract** We performed a systematic analysis of blood DNA methylation profiles from 4483 participants from seven independent cohorts identifying differentially methylated positions (DMPs) associated with psychosis, schizophrenia, and treatment-resistant schizophrenia. Psychosis cases were characterized by significant differences in measures of blood cell proportions and elevated smoking exposure derived from the DNA methylation data, with the largest differences seen in treatment-resistant schizophrenia patients. We implemented a stringent pipeline to meta-analyze epigenome-wide association study (EWAS) results across datasets, identifying 95 DMPs associated with psychosis and 1048 DMPs associated with schizophrenia, with evidence of colocalization to regions nominated by genetic association studies of disease. Many schizophrenia-associated DNA methylation differences were only present in patients with treatment-resistant schizophrenia, potentially reflecting exposure to the atypical antipsychotic clozapine. Our results highlight how DNA methylation data can be leveraged to identify physiological (e.g., differential cell counts) and environmental (e.g., smoking) factors associated with psychosis and molecular biomarkers of treatment-resistant schizophrenia.

## Introduction

Psychosis is a complex and heterogeneous neuropsychiatric condition characterized by a loss of contact with reality, whose symptoms can include delusions and hallucinations. Episodic psychosis and altered cognitive function are major features of schizophrenia, a severe neurodevelopmental disorder that contributes significantly to the global burden of disease (*Whiteford et al., 2013*). Schizophrenia is highly heritable (*Hilker et al., 2018*; *Sullivan et al., 2003*) and recent genetic studies have indicated a complex polygenic architecture involving hundreds of genetic variants that individually confer a minimal increase on the overall risk of developing the disorder (*Purcell et al., 2009*). Large-scale genome-wide association studies (GWAS) have identified approximately 160 regions of the genome harboring common variants robustly associated with the diagnosis of schizophrenia, with evidence for a substantial polygenic component in signals that individually fall below genome-wide levels of significance (*Pardiñas et al., 2018*; *Ripke et al., 2014*). As the majority of schizophrenia-associated variants do not directly index coding changes affecting protein structure, there remains uncertainty about the causal genes involved in disease pathogenesis and how their function is dysregulated (*Maurano et al., 2012*).

A major hypothesis is that GWAS variants predominantly act to influence the regulation of gene expression. This hypothesis is supported by an enrichment of schizophrenia-associated variants in core regulatory domains (e.g., active promotors and enhancers) (*Hannon et al., 2019a*). As a consequence, there has been growing interest in the role of epigenetic variation in the molecular etiology of schizophrenia. DNA methylation is the best-characterized epigenetic modification, acting to influence gene expression via disruption of transcription factor binding and recruitment of methyl-binding proteins that initiate chromatin compaction and gene silencing. Despite being traditionally regarded as a mechanism of transcriptional repression, DNA methylation is actually associated with both increased and decreased gene expression (*Wagner et al., 2014*) and other genomic functions including alternative splicing and promoter usage (*Maunakea et al., 2010*). We previously demonstrated how DNA methylation is under local genetic control (*Hannon et al., 2018a*; *Hannon et al., 2016b*), identifying an enrichment of DNA methylation quantitative trait loci (mQTL) among genomic regions associated with schizophrenia (*Hannon et al., 2016b*). Furthermore, we have used mQTL associations to identify discrete sites of regulatory variation associated with schizophrenia risk variants implicating specific genes within these regions (*Hannon et al., 2016a*; *Hannon et al., 2018a*; *Hannon et al., 2016b*; *Hannon et al., 2017*). Of note, epigenetic variation induced by environmental exposures has been hypothesized as another mechanism by which non-genetic factors can affect risk for neuropsychiatric disorders including schizophrenia (*Dempster et al., 2013*).

The development of standardized assays for quantifying DNA methylation at specific sites across the genome has enabled the systematic analysis of associations between methylomic variation and environmental exposures or diseases (*Murphy and Mill, 2014*). Because DNA methylation is a dynamic process, these epigenome-wide association studies (EWAS) are more complex to design and interpret than GWAS (*Mill and Heijmans, 2013*; *Rakyan et al., 2011*; *Relton and Davey Smith, 2010*). As for observational epidemiological studies of exposures and outcomes, a number of potentially important confounding factors (e.g., tissue or cell type, age, sex, lifestyle exposures, medication, and disorder-associated exposures) that can directly influence DNA methylation need to be considered along with the possibility of reverse causation. Despite these difficulties, recent studies have identified schizophrenia-associated DNA methylation differences in analyses of post-mortem brain tissue (*Jaffe et al., 2016*; *Pidsley et al., 2014*; *Viana et al., 2017*; *Wockner et al., 2014*), and also detected disease-associated variation in peripheral blood samples from both schizophrenia-discordant monozygotic twin pairs (*Dempster et al., 2011*) and clinically ascertained case-control cohorts (*Aberg et al., 2014*; *Hannon et al., 2016a*; *Kinoshita et al., 2014*). We previously reported an EWAS of variable DNA methylation associated with schizophrenia in >1700 individuals, meta-analyzing data from three independent cohorts and identifying methylomic biomarkers of disease (*Hannon et al., 2016a*). Together these data support a role for differential DNA methylation in the molecular etiology of schizophrenia, although it is not clear whether disease-associated methylation differences are themselves secondary to the disorder itself or a result of other schizophrenia-associated factors.

In this study we extend our previous analysis, quantifying DNA methylation across the genome in a total of 4483 participants from seven independent case-control cohorts including patients with

schizophrenia or first-episode psychosis (FEP) (*Figure 1*). This represents the largest EWAS of schizophrenia and psychosis, and one of the largest case-control studies of DNA methylation for any disease phenotype. In each cohort, genomic DNA was isolated from whole blood, and DNA methylation was quantified across the genome using either the Illumina Infinium HumanMethylation450 microarray ('450K array') or the HumanMethylationEPIC microarray ('EPIC array') (see 'Materials and methods'). We implemented a stringent pipeline to meta-analyze EWAS results across datasets to identify associations between psychosis cases and variation in DNA methylation. We show how DNA methylation data can be leveraged to identify biological (e.g., differential cell counts) and environmental (e.g., smoking) factors associated with psychosis, and present evidence for molecular variation associated with clozapine exposure in patients with treatment-resistant schizophrenia (TRS).

## Results

### Study overview and cohort characteristics

We quantified DNA methylation in samples derived from peripheral venous whole blood in seven independent psychosis case-control cohorts (total n = 4483; 2379 cases and 2104 controls). These cohorts represent a range of study designs and recruitment strategies and were initially designed to explore different clinical and etiological aspects of schizophrenia (see Materials and methods and *Table 1*); they include studies of FEP (EU-GEI and IoPPN), established schizophrenia and/or clozapine usage (UCL, Aberdeen, Dublin, IoPPN), mortality in schizophrenia (Sweden), and a study of

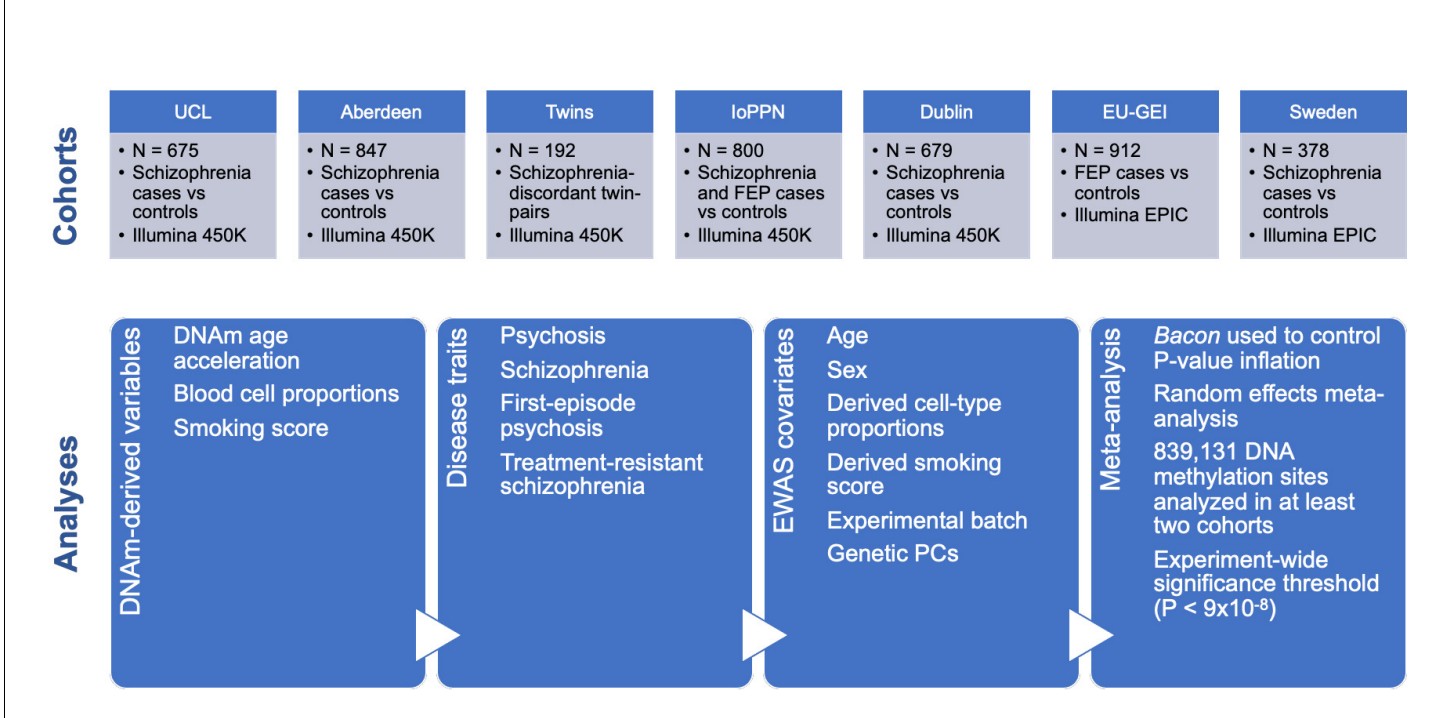

**Figure 1.** Overview of the sample cohorts and analytical approaches used in this study of altered DNA methylation in psychosis and schizophrenia. The online version of this article includes the following figure supplement(s) for figure 1:

**Figure supplement 1.** Forest plot showing the difference in mean age between psychosis cases and controls across each cohort.

**Figure supplement 2.** Scatterplot of the relationship between the first two genetic principal components merged with HapMap phase 3 data for individual cohorts.

**Figure supplement 3.** Scatterplots of DNAmAge derived from the DNA methylation data against actual chronological age for each of the cohorts.

**Figure supplement 4.** Scatterplots of PhenoAge derived from DNA methylation data against actual chronological age for each of the cohorts.

**Figure supplement 5.** Scatterplots of DNAmAge derived from the DNA methylation data against actual chronological age for each of the cohorts.

**Figure supplement 6.** Scatterplots of PhenoAge derived from DNA methylation data against actual chronological age for each of the cohorts.

**Table 1.** Summary of cohort demographics included in the psychosis epigenome-wide association study (EWAS) meta-analysis.

| Cohort | | UCL | Aberdeen | Twins | IoPPN | Dublin | EU-GEI | Sweden | Total |
|---|---|---|---|---|---|---|---|---|---|
| Total sample | | 675 | 847 | 192 | 800 | 679 | 912 | 378 | 4483 |
| % Cases | | 52.3 | 48.9 | 45.3 | 74.6 | 51.3 | 42.9 | 50.0 | 53.1 |
| % Schizophrenia | | 52.3 | 48.9 | 45.3 | 36.3 | 51.3 | 0.0 | 50.0 | 37.5 |
| % First-episode psychosis | | 0.0 | 0.0 | 0.0 | 38.4 | 0.0 | 42.9 | 0.0 | 15.6 |
| % Males | All | 58.7 | 71.1 | 52.1 | 63.0 | 71.0 | 54.4 | 59.5 | 62.6 |
| | Cases | 72.0 | 68.4 | 54.0 | 65.3 | 71.6 | 64.2 | 60.3 | 66.8 |
| | Controls | 44.1 | 73.7 | 50.5 | 56.2 | 70.4 | 47.0 | 58.7 | 57.8 |
| | Chi-square test p-value | 3.81E-13 | 0.103 | 0.730 | 0.024 | 0.804 | 3.68E-07 | 0.834 | 9.35E-10 |
| Age (years) | Mean | 40.4 | 44.6 | 35.3 | 28.8 | 41.7 | 35.3 | 60.0 | 40.5 |
| | SD | 15.0 | 12.9 | 10.8 | 9.46 | 12.0 | 12.8 | 8.86 | 14.7 |
| | Mean in controls | 43.7 | 44.2 | 37.9 | 27.8 | 41.4 | 30.7 | 56.3 | 41.6 |
| | Mean in cases | 36.8 | 44.9 | 33.3 | 30.3 | 42.0 | 38.7 | 63.7 | 39.4 |
| | t-test p-value | 6.55E-09 | 0.529 | 0.033 | 0.007 | 0.505 | 1.24E-22 | 1.05E-16 | |

twins from monozygotic pairs discordant for schizophrenia (Twins). All cohorts were characterized by a higher proportion of male participants (range = 52.1–71.1% male, pooled mean = 62.6% male, *Table 1*) than females. Although there was an overall significantly higher proportion of males among cases compared to controls ($\chi^2$ = 37.5, p=9.35×$10^{-10}$), consistent with reported incidence rates (*Aleman et al., 2003*; *van der Werf et al., 2014*), there was significant heterogeneity in the sex by diagnosis proportions across different cohorts ($\chi^2$ = 348, p=4.86×$10^{-63}$) with the overall excess of male patients driven by two cohorts (UCL [$\chi^2$ = 52.7, p=3.81×$10^{-13}$] and EU-GEI [$\chi^2$ = 25.9, p=3.68×$10^{-7}$]). Most cohorts were enriched for young and middle-aged adults, although there was considerable heterogeneity across the studies reflecting the differing sampling strategies (*Table 1*). For example, the IoPPN cohort has the lowest average age, reflecting the inclusion of a large number of FEP patients (mean = 34.9 years; SD = 12.42 years) (*Di Forti et al., 2009*). In contrast, individuals in the Sweden cohort were older (mean = 60.0 years; SD = 8.9 years) (*Kowalec et al., 2019*). There was no overall difference in mean age between cases and controls (mean difference = 0.076 years; p=0.975) (*Figure 1—figure supplement 1*), although differences were apparent in individual cohorts; in UCL (mean difference = 6.8 years; p=6.55×$10^{-9}$) and IoPPN (mean difference = 6.2 years; p=1.46×$10^{-11}$), patients were significantly older than controls, while in the EU-GEI (mean difference = −7.9 years; p=1.24×$10^{-22}$) and the Sweden cohort (mean difference = −7.3 years; p=1.05×$10^{-16}$), the cases were significantly younger. With the exception of individuals in the IoPPN and EU-GEI cohorts, which are more ethnically diverse, individuals included in this study were predominantly Caucasian. SNP array data from each donor was merged with HapMap phase 3 data, and genetic principal components (PCs) were calculated with GCTA (*Yang et al., 2011*) to further confirm the ethnicity of each sample (*Figure 1—figure supplement 2*).

## Psychosis patients are characterized by differential blood cell proportions and smoking levels using measures derived from DNA methylation data

A number of robust statistical classifiers have been developed to derive estimates of both biological phenotypes (e.g., age [*Hannum et al., 2013*; *Horvath, 2013*; *Zhang et al., 2019*] and the proportion of different blood cell types in a whole blood sample [*Houseman et al., 2012*; *Koestler et al., 2013*]) and environmental exposures (e.g., tobacco smoking [*Elliott et al., 2014*; *Sugden et al., 2019*]) from DNA methylation data. These estimates can be used to identify differences between groups and are often included as covariates in EWAS analyses where empirically measured data is not available. For each individual included in this study, we calculated two measures of 'epigenetic age' from the DNA methylation data; DNAmAge using the *Horvath, 2013* multitissue clock, which was developed to predict chronological age, and PhenoAge, which was developed as biomarker of

advanced biological aging (*Levine et al., 2018*). We found a strong correlation between reported age and both derived age estimates across the cohorts (Pearson's correlation coefficient range 0.821–0.928 for DNAmAge and 0.795–0.910 for PhenoAge) and no evidence for age acceleration – that is, the difference between epigenetic age and chronological age – between patients with psychosis and controls (*Kowalec et al., 2019*; *Figure 1—figure supplements 3* and *4*).

Because of the importance of considering variation in the composition of the constituent cell types in analyses of complex cellular mixtures (*Mill and Heijmans, 2013*; *Relton and Davey Smith, 2010*), we used established methods to estimate the proportion (*Houseman et al., 2012*; *Koestler et al., 2013*) and abundance (*Horvath, 2013*) of specific cell types in whole blood. Using a random effects meta-analysis to combine the results across the seven cohorts, which were adjusted for age, sex, and DNAm smoking score, we found that psychosis cases had elevated estimated proportions of granulocytes (mean difference = 0.0431; p=$5.09\times10^{-4}$) and monocytes (mean difference = 0.00320; p=$1.15\times10^{-4}$), and significantly lower proportions of CD4$^+$ T-cells (mean difference = −0.0177; p=0.00144), CD8$^+$ T-cells (mean difference = −0.0144; p=0.00159), and natural killer cells (mean difference = −0.0113; p=0.00322) (*Table 2* and *Figure 2*). Interestingly, the differences in granulocytes, natural killer cells, CD4$^+$ T-cells, and CD8$^+$ T-cells were most apparent in cohorts comprising patients with a diagnosis of schizophrenia (*Figure 2*), with cohorts including FEP patients characterized by weaker or null effects. Limiting the analysis of derived blood cell estimates to a comparison of schizophrenia cases and controls did not perceivably change the estimated differences of our random effects model but did reduce the magnitude of heterogeneity compared to including the FEP cases (*Supplementary file 1*). This indicates that changes in blood cell proportions may reflect a consequence of diagnosis, reflecting the fact that people with schizophrenia are likely to have been exposed to a variety of medications, social adversities, and somatic ill-health – and for longer periods – than FEP patients. Finally, we used an established algorithm to derive a quantitative DNA methylation 'smoking score' for each individual (*Elliott et al., 2014*), building on our previous work demonstrating the utility of this variable for characterizing differences in smoking exposure between schizophrenia patients and controls, and using it as a covariate in an EWAS (*Hannon et al., 2016a*). We observed a significantly increased DNA methylation smoking score (*Figure 3*) in psychosis patients compared to controls across all cohorts (mean difference = 3.89; p=$2.88\times10^{-11}$). Although of smaller effect, this difference was also present when comparing FEP and controls in the EU-GEI cohort (mean difference = 2.38; p=$2.68\times10^{-8}$). As expected, for individuals where self-reported smoking data was available, the DNA methylation smoking score was significantly elevated in current and former smokers compared to never smokers (*Figure 3—figure supplement 1*).

**Table 2.** Results of a meta-analysis of differences in blood cell composition estimates derived from DNA methylation data between schizophrenia cases and controls.

| Cell type | Measure type | Number of cohorts | Random effects model | | | Fixed effects model | | | Heterogeneity p-Value |
|---|---|---|---|---|---|---|---|---|---|
| | | | Mean difference | SE | p-Value | Mean difference | SE | p-Value | |
| Monocytes | Proportion | 7 | 0.00320 | 0.00083 | 0.000115 | 0.00320 | 0.00083 | 0.000115 | 0.6490 |
| Granulocytes | Proportion | 7 | 0.04312 | 0.01241 | 0.000509 | 0.03930 | 0.00315 | 1.21E-35 | 2.22E-16 |
| Natural killer cells | Proportion | 7 | −0.01135 | 0.00385 | 0.003221 | −0.00827 | 0.00133 | 4.48E-10 | 2.43E-08 |
| CD4$^+$ T-cells | Proportion | 7 | −0.01767 | 0.00555 | 0.00144 | −0.01569 | 0.00196 | 1.15E-15 | 1.23E-07 |
| CD8$^+$ T-cells | Proportion | 7 | −0.01444 | 0.00457 | 0.001586 | −0.01443 | 0.00148 | 1.31E-22 | 8.13E-10 |
| B-cells | Proportion | 7 | −0.00495 | 0.00280 | 0.077103 | −0.00477 | 0.00102 | 2.75E-06 | 2.25E-07 |
| PlasmaBlast | Abundance | 5 | 0.05626 | 0.02987 | 0.059671 | 0.05332 | 0.00722 | 1.55E-13 | 8.45E-13 |
| CD8pCD28nCD45RAn | Abundance | 5 | 0.06280 | 0.22674 | 0.781792 | 0.10797 | 0.14981 | 0.4711 | 0.0826 |
| CD8 naive T-cells | Abundance | 5 | 7.21687 | 3.12594 | 0.02096 | 8.03957 | 1.89169 | 2.14E-05 | 0.0443 |
| CD4 naive T-cells | Abundance | 5 | 11.77240 | 4.72532 | 0.012726 | 11.77240 | 4.72532 | 0.0127 | 0.824 |

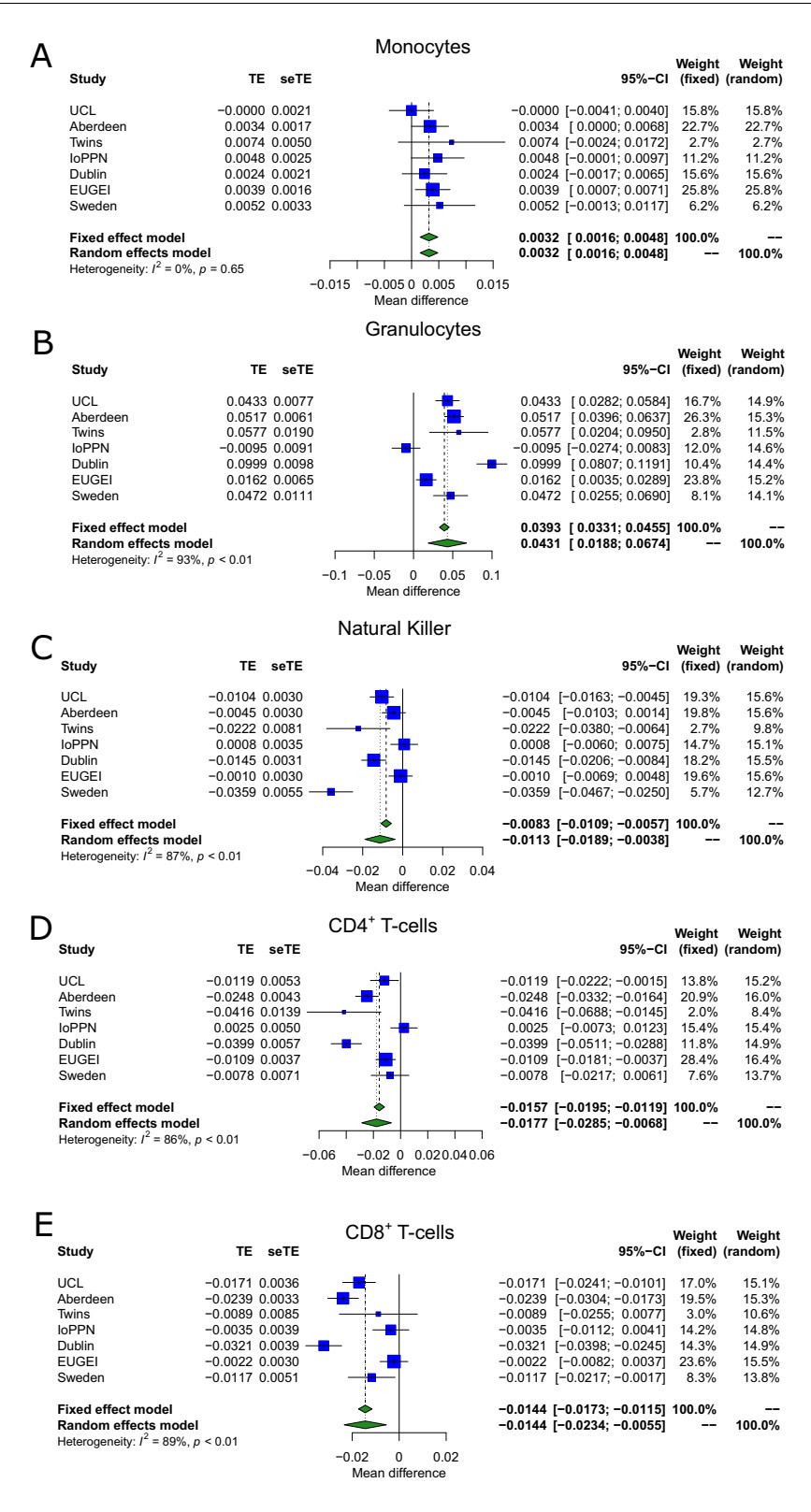

**Figure 2.** Blood cell-type proportions derived from DNA methylation data are altered in psychosis. Shown are forest plots from meta-analyses of differences in blood cell proportions derived from DNA methylation data between psychosis patients and controls for (A) monocytes, (B) granulocytes, (C) natural killer cells, (D) CD4+ T-cells, and (E) CD8+ T-cells. TE: treatment effect (i.e., the mean difference between cases and controls); seTE: standard error of the treatment effect.

*Figure 2 continued on next page*

*Figure 2 continued*

The online version of this article includes the following figure supplement(s) for figure 2:

**Figure supplement 1.** Treatment-resistant schizophrenia patients prescribed clozapine are characterized by altered blood cell proportions.

**Figure supplement 2.** Additive effect of schizophrenia and treatment resistance on granulocyte proportions.

**Figure supplement 3.** Additive effect of schizophrenia and treatment resistance on CD8$^+$ T-cell proportions.

## An EWAS meta-analysis identifies DNA methylation differences associated with psychosis

To identify differentially methylated positions (DMPs) in blood associated with psychosis, we performed an association analysis within each of the seven schizophrenia and FEP cohorts controlling for age, sex, derived cellular composition variables (from DNA methylation data), derived smoking score (from DNA methylation data), and experimental batch (see 'Materials and methods'). We used a Bayesian method to control p-value inflation using the R package *bacon* (*van Iterson et al., 2017*) before combining the estimated effect sizes and standard errors across cohorts with a random effects meta-analysis, including all autosomal and X-chromosome DNA methylation sites analyzed in at least two cohorts (n = 839,131 DNA methylation sites) (see 'Materials and methods'). Using an experiment-wide significance threshold derived for the Illumina EPIC array (*Mansell et al., 2019*) (p<9×10$^{-8}$), we identified 95 psychosis-associated DMPs mapping to 93 independent loci and annotated to 68 genes (*Figure 4A* and *Supplementary file 1*). Across these DMPs, the mean difference in DNA methylation between cases and controls was relatively small (0.789%, SD = 0.226%) and there was a striking enrichment of hypermethylated DMPs in psychosis cases (n = 91 DMPs [95.8%] hypermethylated; p=1.68×10$^{-22}$). A number of the top-ranked DMPs are annotated to genes that have direct relevance to the etiology of psychosis including the GABA transporter *SLC6A12* (*Park et al., 2011*) (cg00517261, mean difference = 0.663%; p=1.53×10$^{-8}$), the GABA receptor

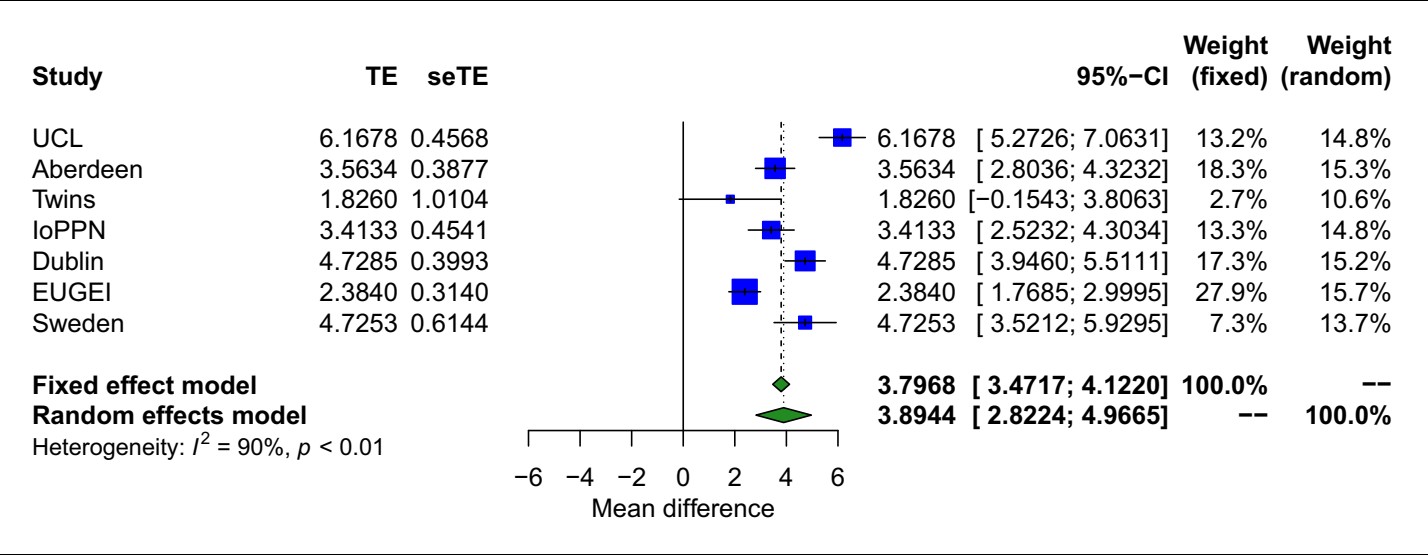

**Figure 3.** Smoking scores derived from DNA methylation data highlight that psychosis patients are characterized by an elevated exposure to tobacco smoking. Forest plot from a meta-analysis of differences in smoking score derived from DNA methylation data between psychosis patients and controls. The smoking score was calculated from DNA methylation data using the method described by *Elliott et al., 2014*. TE: treatment effect (i.e., the mean difference between cases and controls); seTE: standard error of the treatment effect.

The online version of this article includes the following figure supplement(s) for figure 3:

**Figure supplement 1.** Current and former smokers are characterized by a significantly higher smoking score derived from DNA methylation data than non-smokers.

**Figure supplement 2.** Treatment-resistant schizophrenia is associated with significantly higher DNA methylation-derived smoking scores.

**Figure supplement 3.** Treatment-resistant schizophrenia (TRS) patients show an elevated exposure to tobacco smoking relative to non-TRS and controls in a model testing both schizophrenia diagnosis status and TRS status simultaneously.

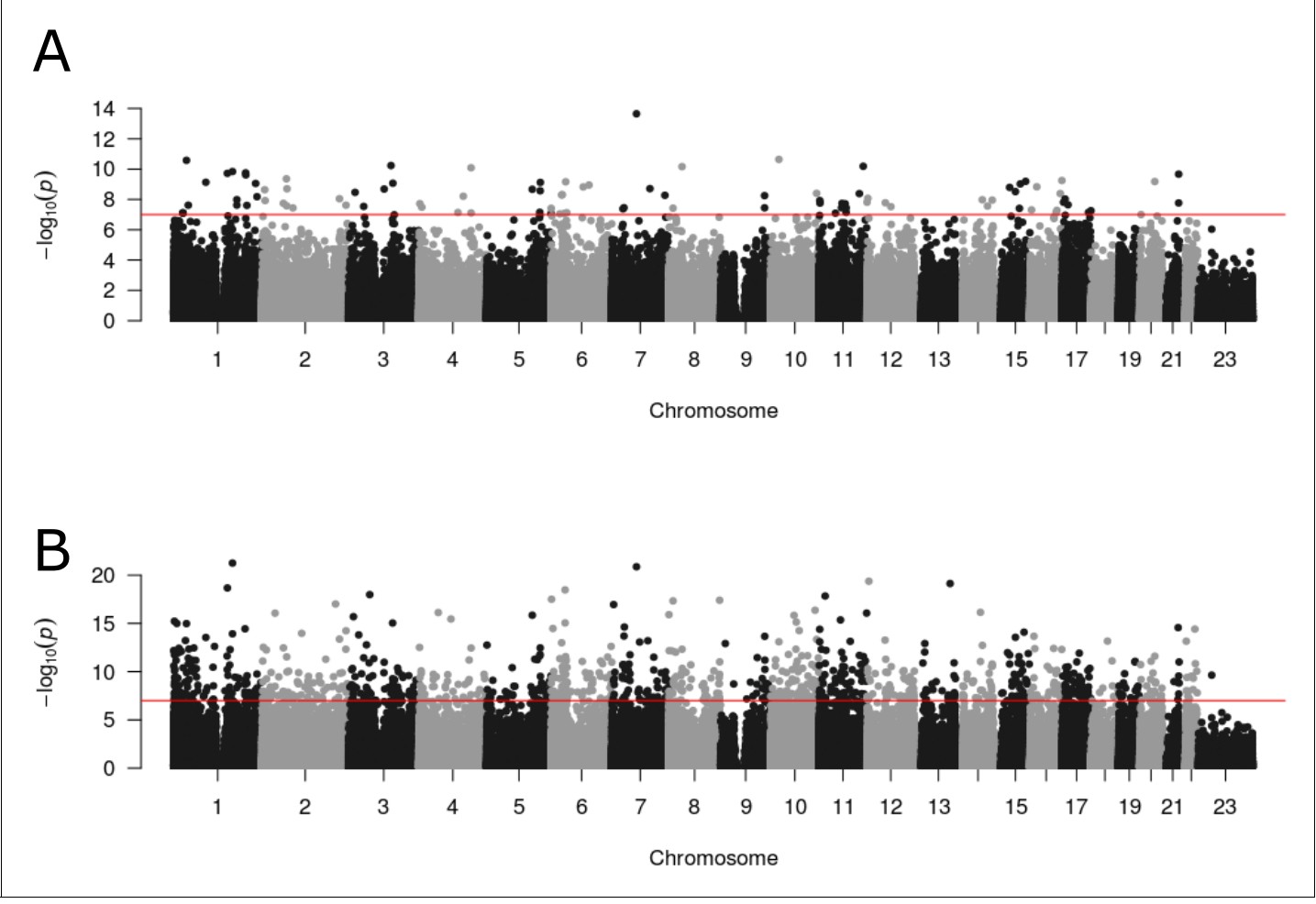

**Figure 4.** Differential DNA methylation at multiple loci across the genome is associated with psychosis and schizophrenia. Manhattan plots depicting the $-\log_{10}$ p-value from the epigenome-wide association study meta-analysis (y-axis) against genomic location (x-axis). (**A**) presents results from the analysis comparing psychosis patients and controls, and (**B**) presents results from the analysis comparing diagnosed schizophrenia cases and controls. The online version of this article includes the following figure supplement(s) for figure 4:

**Figure supplement 1.** Including genetic principal components (PCs) into DNA methylation analysis models has little effect on the results in ethnically heterogeneous cohorts.

*GABBR1* (*Le-Niculescu et al., 2007*) (cg00667298, mean difference = 0.619%; p=5.07×10⁻⁹), and the calcium voltage-gated channel subunit gene *CACNA1C* (cg01833890, mean difference = 0.458%; p=8.42×10⁻⁹) that is strongly associated with schizophrenia and bipolar disorder (*Cross-Disorder Group of the Psychiatric Genomics Consortium, 2013*; *Psychiatric GWAS Consortium Bipolar Disorder Working Group, 2011*; *Ripke et al., 2011*; *Figure 5*).

## A specific focus on clinically diagnosed schizophrenia cases identifies more widespread DNA methylation differences

We next repeated the EWAS focusing specifically on the subset of psychosis cases with diagnosed schizophrenia (schizophrenia cases = 1681, controls = 1583). Compared to our EWAS of psychosis, we identified more widespread differences in DNA methylation (*Figure 4B*), with 1048 schizophrenia-associated DMPs (p<9×10⁻⁸) representing 1013 loci and annotated to 692 genes (*Supplementary file 1*). Although the list of schizophrenia-associated DMPs included 61 (64.21%) of the psychosis-associated DMPs, the total number of significant differences was much larger, potentially reflecting the less heterogeneous clinical characteristics of the cases. Schizophrenia-associated DMPs had a mean difference of 0.789% (SD = 0.204%) and, like the psychosis-associated

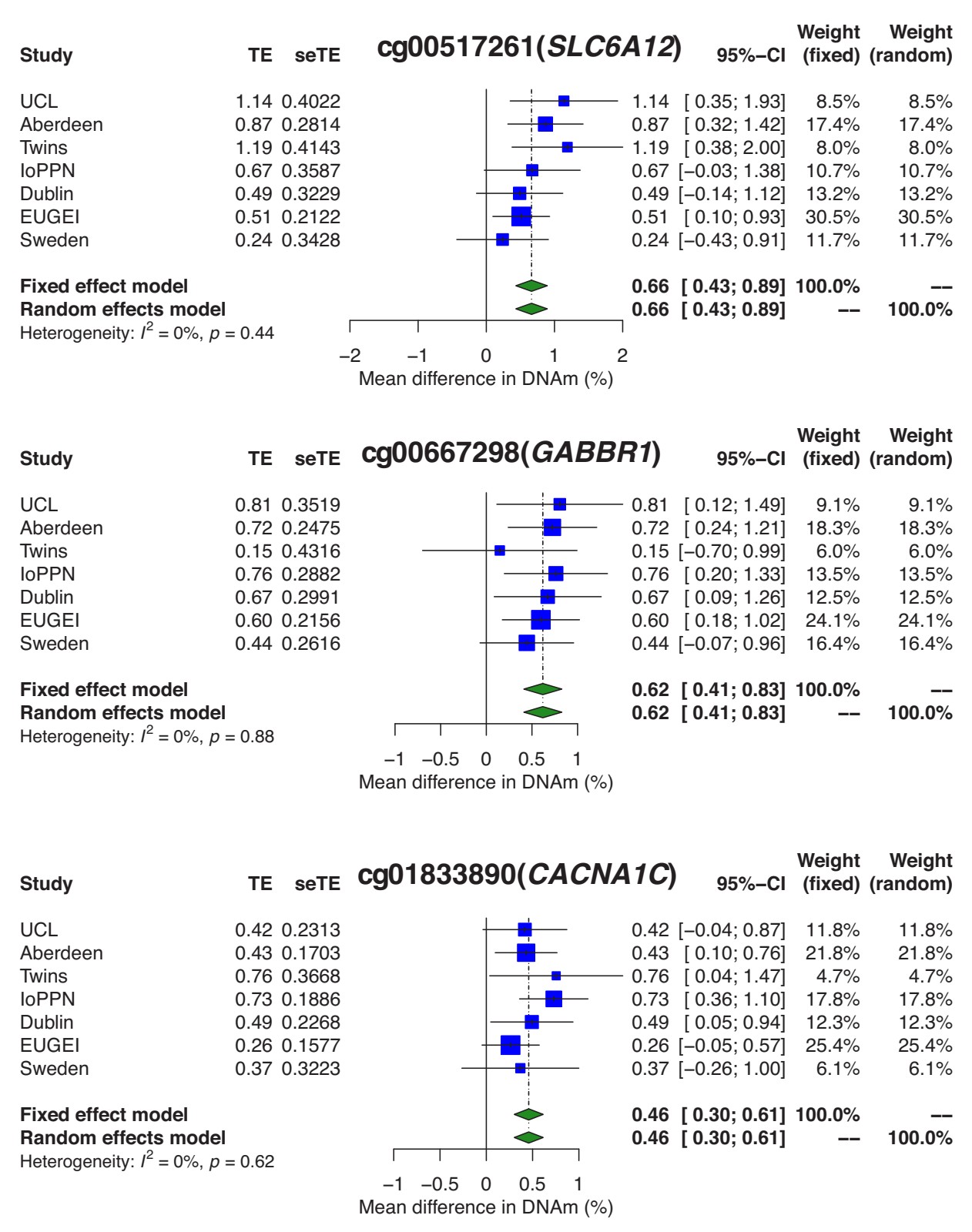

**Figure 5.** Psychosis-associated differential DNA methylation at sites annotated to genes previously implicated in disease etiology. Shown are forest plots for differentially methylated positions (DMPs) annotated to the GABA transporter *SLC6A12* (cg00517261; p=1.53×10$^{-8}$), the GABA receptor *GABBR1* (cg00667298; p=5.07×10$^{-9}$), and the calcium voltage-gated channel subunit gene *CACNA1C* (cg01833890; p=8.42×10$^{-9}$). TE: treatment effect (i.e., the mean difference between cases and controls); seTE: standard error of the treatment effect.

differences, were significantly enriched for sites that were hypermethylated in cases compared to controls (n = 897 [87.4%]; p=$1.27 \times 10^{-129}$). A number of the top-ranked DMPs are annotated to genes that have direct relevance to the etiology of schizophrenia and gene ontology (GO) analysis highlighted multiple pathways previously implicated in schizophrenia including several related to the extracellular matrix (*Berretta, 2012*) and cell-cell adhesion (*O'Dushlaine et al., 2011*; *Supplementary file 1*). Given the large range of ages across the samples included in this study, we tested whether there was evidence for a relationship between age and differential DNA methylation at the 1048 schizophrenia DMPs by refitting our analysis model using an additional interaction term between age and schizophrenia status individually for each cohort prior to the interaction effects being meta-analyzed (see 'Materials and methods'). Overall, we found limited evidence for a relationship between age and DNA methylation at schizophrenia-associated DMPs; controlling for multiple testing (p<0.00004771), only two (0.002%) DMPs were identified as showing a significant interaction with age (*Supplementary file 1*). We used the same approach to explore for an interaction between sex and DNA methylation, finding no evidence for sex differences at these sites or evidence for a significant interaction between sex and DNA methylation (p<0.00004771) (*Supplementary file 1*). Finally, although most of the cohorts included in this study were predominantly Caucasian, there was some ethnic heterogeneity in the IoPPN and EU-GEI cohorts. To explore the extent to which this diversity might be influencing our results, we merged SNP array data from each donor with HapMap phase 3 data and calculated genetic PCs using GCTA (*Yang et al., 2011*; *Figure 1—figure supplement 2*). We re-analyzed data from individual cohorts including increasing numbers of genetic PCs to the model, finding that even in the most ethnically diverse cohort (IoPPN) the inclusion of up to five genetic PCs had negligible effects, with a very strong correlation in test statistics between models (*Figure 4—figure supplement 1*).

## Schizophrenia-associated DNA methylation differences show overlap with previous analyses of schizophrenia and other traits

Two of our experiment-wide significant schizophrenia-associated DMPs (cg00390724 and cg09868768) overlapped with those reported in a previous smaller whole blood schizophrenia EWAS performed by *Montano et al., 2016* with the same direction of effect; of note, 119 (71.3%) of the 167 replicated DMPs reported by this study were characterized by a consistent direction of effect in our meta-analysis, representing a significantly higher rate than expected by chance (p=$3.83 \times 10^{-8}$). Unfortunately, we could not check the extent to which our schizophrenia-associated DMPs were replicated in the Montano et al. dataset because the full results from their analysis are not publicly available. We next compared our results with those from a prefrontal cortex EWAS meta-analysis of schizophrenia also performed by our group (*Viana et al., 2017*), finding that 627 (60.2%) of the 1042 DMPs tested in both analyses had the same direction of effect, a significantly higher rate than expected by chance (p=$5.43 \times 10^{-11}$). Finally, we also explored the extent to which DMPs associated with schizophrenia overlapped with other traits using the database of results in the online EWAS catalog (http://ewascatalog.org/); across EWAS undertaken using blood DNA (isolated from whole blood or cord blood), this resource includes 101,091 significant DMPs (at p<$1 \times 10^{-7}$) associated with 87 traits. Of the 1048 schizophrenia-associated DMPs identified in our meta-analysis, 219 (20.9%) were present in the database and significantly associated with 18 different traits (*Supplementary file 1*). Where effect sizes for individual DMPs were available in the EWAS catalog, we tested for an enrichment of consistent (or discordant) associations to those identified with schizophrenia. Schizophrenia DMPs also associated with C-reactive protein (CRP) and gestational age, for example, were significantly enriched for a consistent direction of effect (CRP: 10 overlapping DMPs, 10 consistent direction of effect, p=0.001953; gestational age: 105 overlapping DMPs, 72 consistent direction of effect, p=0.000178). In contrast, schizophrenia DMPs also associated with age and high-density lipoprotein (HDL) cholesterol were enriched for discordant effect directions (age: 30 overlapping DMPs, 28 same direction of effect, p=$8.68 \times 10^{-7}$; HDL: 12 overlapping DMPs, 12 same direction of effect, p=0.00049) (*Figure 6*).

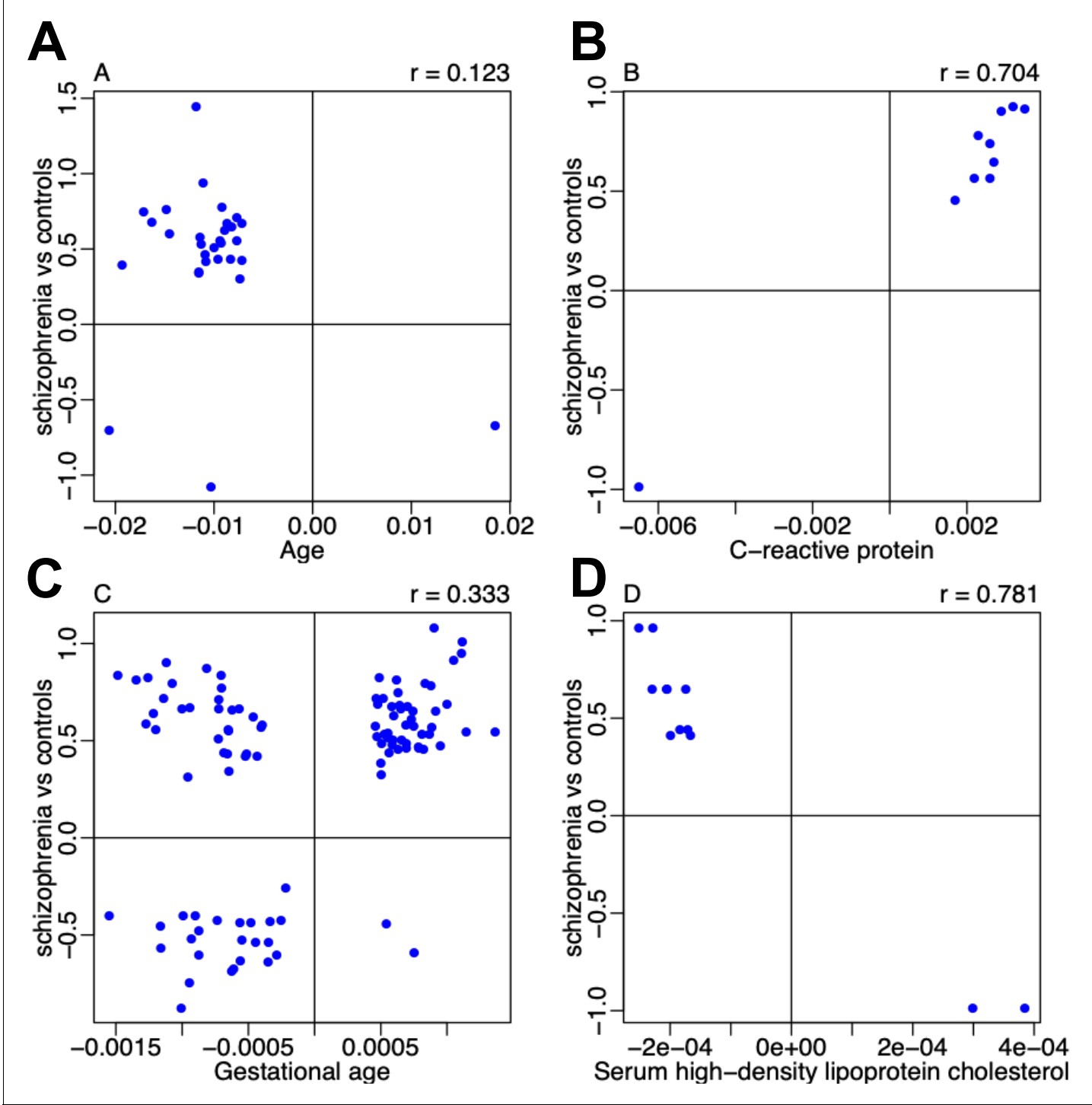

**Figure 6.** Comparison of effect sizes for schizophrenia-associated differentially methylated positions (DMPs) overlapping with epigenome-wide association study (EWAS) results for other traits. Shown for each overlapping DMP is the association effect size for the other trait (x-axis) taken from the online EWAS catalog (http://ewascatalog.org/) compared to the effect size identified in our meta-analysis of schizophrenia (y-axis).

## Schizophrenia-associated DMPs colocalize to regions nominated by genetic association studies

As the etiology of schizophrenia has a large genetic component, we next sought to explore the extent to which DNA methylation at schizophrenia-associated DMPs is influenced by genetic variation. Using results from a quantitative genetic analysis of DNA methylation in monozygotic and dizygotic twins (*Hannon et al., 2018c*), we found that DNA methylation at schizophrenia-associated DMPs is more strongly influenced by additive genetic factors compared to non-associated sites matched for comparable means and standard deviations (*Figure 7*) (mean additive genetic component across DMPs = 23.0%; SD = 16.8%; $p=1.61\times10^{-87}$). Using a database of blood DNA mQTL previously generated by our group (*Hannon et al., 2018a*), we identified common genetic variants associated with 256 (24.4%) of the schizophrenia-associated DMPs. Across these 256 schizophrenia-associated DMPs, there were a total of 455 independent genetic associations with 448 genetic variants, indicating that some of these DMPs are under polygenic control with multiple genetic variants associated. Of note, 31 of these genetic variants are located within 12 schizophrenia-associated GWAS regions (*Supplementary file 1*) with 19 genetic variants associated with schizophrenia DMPs located in the MHC region on chromosome 6. To further support an overlap between GWAS and EWAS signals for schizophrenia, we compared the list of genes identified in this study with those from the largest GWAS meta-analysis of schizophrenia (*Pardiñas et al., 2018*) identifying 21 schizophrenia-associated DMPs located in 11 different GWAS regions. To more formally test for an enrichment of differential DNA methylation across schizophrenia-associated GWAS regions, we calculated a combined EWAS p-value for each of the GWAS-associated regions using all DNA methylation sites within each region identifying 21 significant regions ($p<3.16\times10^{-4}$, corrected for testing 158 regions; *Supplementary file 1*). Three of these regions also contained a significant schizophrenia-associated DMP and a genetic variant associated with that schizophrenia-associated DMP. These include a region located within the MHC, another located on chromosome 17 containing *DLG2*, *TOM1L2,* and overlapping the Smith-Magenis syndrome deletion, and another on chromosome 16 containing *CENPT* and *PRMT7*.

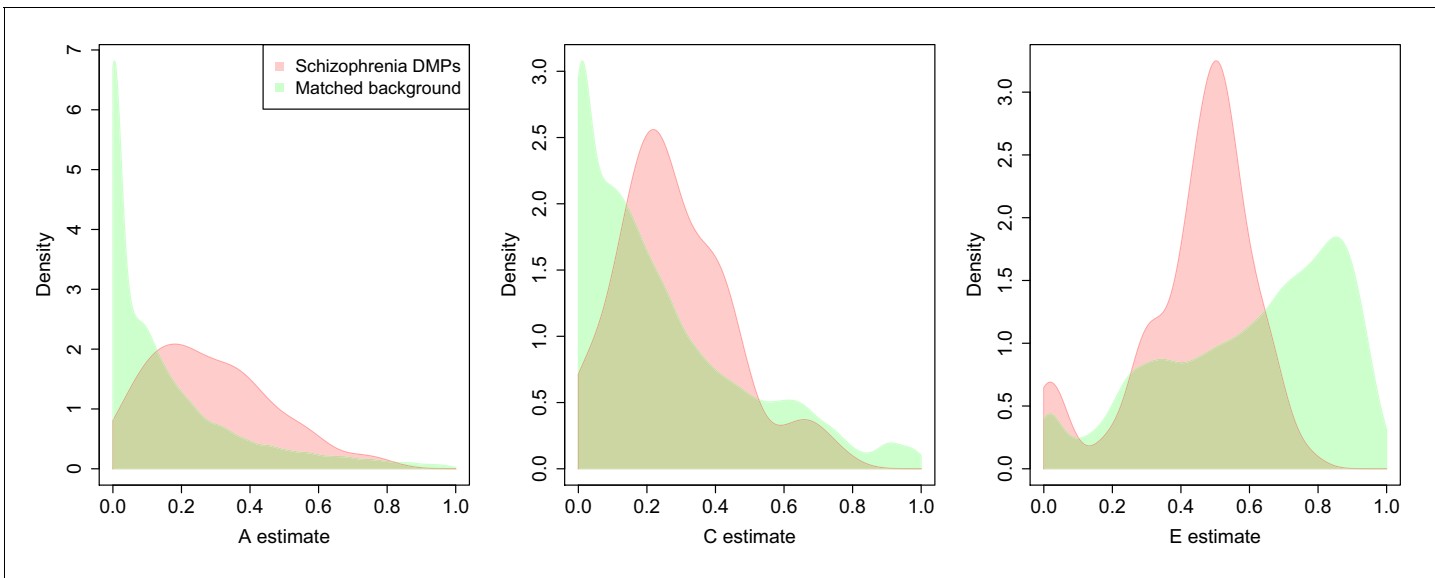

**Figure 7.** DNA methylation at sites associated with schizophrenia is more strongly influenced by genetic factors and common environmental influences than equivalent matched sites across the genome. A series of density plots for estimates of additive genetic effects (A, left), common environmental effects (C, middle), and non-shared environmental effects (E, right) derived using data from a dataset generated by *Hannon et al., 2018b*: schizophrenia DMPs (red) and matched background sites (green).

## Schizophrenia-associated patterns of DNA methylation are observed in individuals with FEP

To explore whether schizophrenia-associated differences in DNA methylation are present before a formal diagnosis of schizophrenia, we next performed an EWAS of FEP in the IoPPN and EU-GEI cohorts (total n = 698 FEP cases and 724 controls), meta-analyzing the results across 384,217 common DNAm sites. Although we identified no significant DMPs at our stringent experiment-wide significance threshold, this is not surprising given the greatly attenuated sample size and the high phenotypic heterogeneity among individuals with FEP compared to diagnosed schizophrenia; both factors negatively influence power to detect effects. We next repeated our EWAS of diagnosed schizophrenia, excluding the IoPPN cohort to ensure that there were no overlapping samples between the schizophrenia vs. control analysis and the FEP vs. control analysis, identifying 125 significant DMPs of which 101 were also tested in the FEP EWAS. To see if there was any evidence for differential DNAm at these sites prior to a diagnosis of schizophrenia, we compared the estimated differences between schizophrenia cases and controls and FEP cases and controls (*Supplementary file 1*). Strikingly, 96 (95.0%) of the tested DMPs had a consistent direction of effect in the FEP EWAS, a significantly higher rate than expected by chance (p=$6.58{\times}10^{-23}$). While this result is consistent with schizophrenia-associated differences being present prior to diagnosis, it is not sufficient to state that they are causal; they may still reflect some underlying environmental risk factor or be a consequence of FEP (e.g., medication exposure).

## TRS cases differ from treatment-responsive schizophrenia patients for blood cell proportion estimates and smoking score derived from DNA methylation data

Up to 25% of schizophrenia patients are resistant to the most commonly prescribed antipsychotic medications, and clozapine is a second-generation antipsychotic often prescribed to patients with such TRS who may represent a more severe subgroup (*Ajnakina et al., 2018*). Using data from four cohorts for which medication records were available (UCL, Aberdeen, IoPPN, and Sweden), we performed a within-schizophrenia analysis comparing schizophrenia patients prescribed clozapine (described as TRS cases) and those prescribed standard antipsychotic medications (total n = 399 TRS and 636 non-TRS). Across each of the four cohorts, the proportion of males prescribed clozapine was slightly higher than the proportion of males on other medications ($\chi^2$ = 7.04; p=$7.96{\times}10^{-3}$; *Supplementary file 1*) consistent with findings from epidemiological studies that report increased rates of clozapine prescription in males (*Bachmann et al., 2017*), although there was statistically significant heterogeneity in the sex distribution between groups across cohorts ($\chi^2$ = 20.5; p=0.0150). TRS cases were significantly younger than non-TRS cases (mean difference = −5.48 years; p=0.00533), although there was significant heterogeneity between the cohorts ($I^2$ = 89%; p=$7.40{\times}10^{-32}$). There was no evidence of accelerated epigenetic aging between TRS and non-TRS patients (*Figure 1—figure supplement 5* and *Figure 1—figure supplement 6*). Interestingly, cellular composition variables derived from the DNA methylation data suggest that TRS cases are characterized by a significantly higher proportion of granulocytes (meta-analysis mean difference = 0.00283; p=$8.10{\times}10^{-6}$) and lower proportions of CD8$^+$ T-cells (mean difference = −0.0115; p=$4.37{\times}10^{-5}$; *Supplementary file 1* and *Figure 2—figure supplement 1*) compared to non-TRS cases. Given the finding of higher derived granulocyte and lower CD8$^+$ T-cell levels in the combined psychosis patient group compared to controls described above, a finding driven primarily by patients with schizophrenia, we performed a multiple regression analysis of granulocyte proportion to partition the effects associated with schizophrenia status from effects associated with TRS status. After including a covariate for TRS, schizophrenia status was not significantly associated with granulocyte proportion using a random effects model (p=0.210) but there was significant heterogeneity of effects across the four cohorts ($I^2$ = 91%, p=$4.93{\times}10^{-7}$). Within the group of patients with schizophrenia, however, there were notable differences between TRS and non-TRS groups (mean difference = 0.0275; p=$3.22{\times}10^{-6}$; *Figure 2—figure supplement 2*). In contrast, a multiple regression analysis found that both schizophrenia status (mean difference = −0.0113; p=0.00818) and TRS status (mean difference = −0.0116; p=$2.82{\times}10^{-5}$) had independent additive effects on CD8$^+$ T-cell proportion (*Figure 2—figure supplement 3*). Finally, TRS was also associated with significantly higher DNA methylation-derived smoking scores than non-TRS in all four cohorts

(mean difference = 2.16; p=7.79×10$^{-5}$; *Figure 3—figure supplement 2*). Testing both schizophrenia diagnosis status and TRS status simultaneously, we found that both remained significant; schizophrenia diagnosis was associated with a significant increase in smoking score (mean difference = 3.98; p=2.19×10$^{-8}$) with TRS status associated with an additional increase within cases (mean difference = 2.15; p=2.22×10$^{-7}$) (*Figure 3—figure supplement 3*).

## There are widespread DMPs between TRS patients and treatment-responsive patients

We next performed an EWAS within schizophrenia patients comparing TRS cases to non-TRS cases, including each autosomal and X-chromosome DNA methylation site analyzed in at least two cohorts (n = 431,659 DNA methylation sites). We identified seven DMPs associated with clozapine exposure (p<9×10$^{-8}$; *Supplementary file 1*) with a mean difference of 1.47% (SD = 0.242%) and all sites being characterized by elevated DNA methylation in TRS cases (p=0.0156). We were interested in whether the DNA methylation differences associated with TRS overlapped with those identified between all schizophrenia cases and non-psychiatric controls. Although there was no direct overlap between the clozapine associated DMPs and the schizophrenia-associated DMPs identified for each analysis, the direction of effects across the 1048 schizophrenia-associated DMPs were enriched for consistent effects (n = 738 [70.4%] DMPs with consistent direction; p=7.57×10$^{-41}$). Given these observations, we formally tested whether the schizophrenia-associated differences are driven by the subset of TRS cases on clozapine by fitting a model that simultaneously estimates the effect of schizophrenia status and TRS status across all 1048 sites (*Supplementary file 1*). While the vast majority of schizophrenia-associated DMPs remained at least nominally significant (n = 1003, 95.7%; p<0.05) between schizophrenia patients and controls; among those that did not, 25 (2.39%) had a significant effect associated with TRS status. For example, differential DNA methylation at the schizophrenia-associated DMP cg16322565, located in the *NR1L2* gene on chromosome 3 (schizophrenia EWAS meta-analysis: mean DNA methylation difference = 0.907%; p=3.52×10$^{-9}$), is driven primarily by cases with TRS (*Figure 8*; multiple regression analysis mean DNA methylation difference between schizophrenia cases and controls = 0.323%; p=0.123; mean DNA methylation difference between TRS cases and non-TRS controls = 1.01%; p=8.71×10$^{-5}$). One hundred and fifty-two (14.5%) of the schizophrenia-associated DMPs were associated with a significant effect between schizophrenia cases and controls and a significant effect within schizophrenia patients between TRS and non-TRS patients, with the majority (128 [84.2%]) characterized by the same direction of effect in both groups and indicative of an additive effect of both schizophrenia diagnosis and TRS status (e.g., *Figure 8—figure supplement 1*). Of particular interest are 24 DMPs which are significantly associated with both schizophrenia and TRS but with an opposite direction of effect, highlighting how that at some DNA methylation sites, TRS counteracts changes induced by schizophrenia (e.g., *Figure 8—figure supplement 2*). Taken together, 177 (16.9%) of the schizophrenia-associated DMPs identified in our EWAS meta-analysis are influenced by TRS and reflect either differences induced by exposure to a specific antipsychotic therapy or other differences (e.g., treatment resistance) in individuals who are prescribed clozapine.

## Discussion

We report the most comprehensive study of methylomic variation associated with psychosis and schizophrenia, profiling DNA methylation across the genome in peripheral blood samples from 2379 cases and 2104 controls. We show how DNA methylation data can be leveraged to derive measures of blood cell counts and smoking that are associated with psychosis. Using a stringent pipeline to meta-analyze EWAS results across datasets, we identify novel DMPs associated with both psychosis and a more refined diagnosis of schizophrenia. Of note, we show evidence for the colocalization of genetic associations for schizophrenia and differential DNA methylation. Finally, we present evidence for differential methylation associated with TRS, potentially reflecting differences in DNA methylation associated with exposure to the atypical antipsychotic drug clozapine.

We identify robust psychosis-associated differences in cellular composition estimates derived from DNA methylation data, with cases having increased proportions of monocytes and granulocytes and decreased proportions of natural killer cells, CD4$^{+}$ T-cells and CD8$^{+}$ T-cells, compared to non-psychiatric controls. This analysis extends previous work based on a subset of these data, which

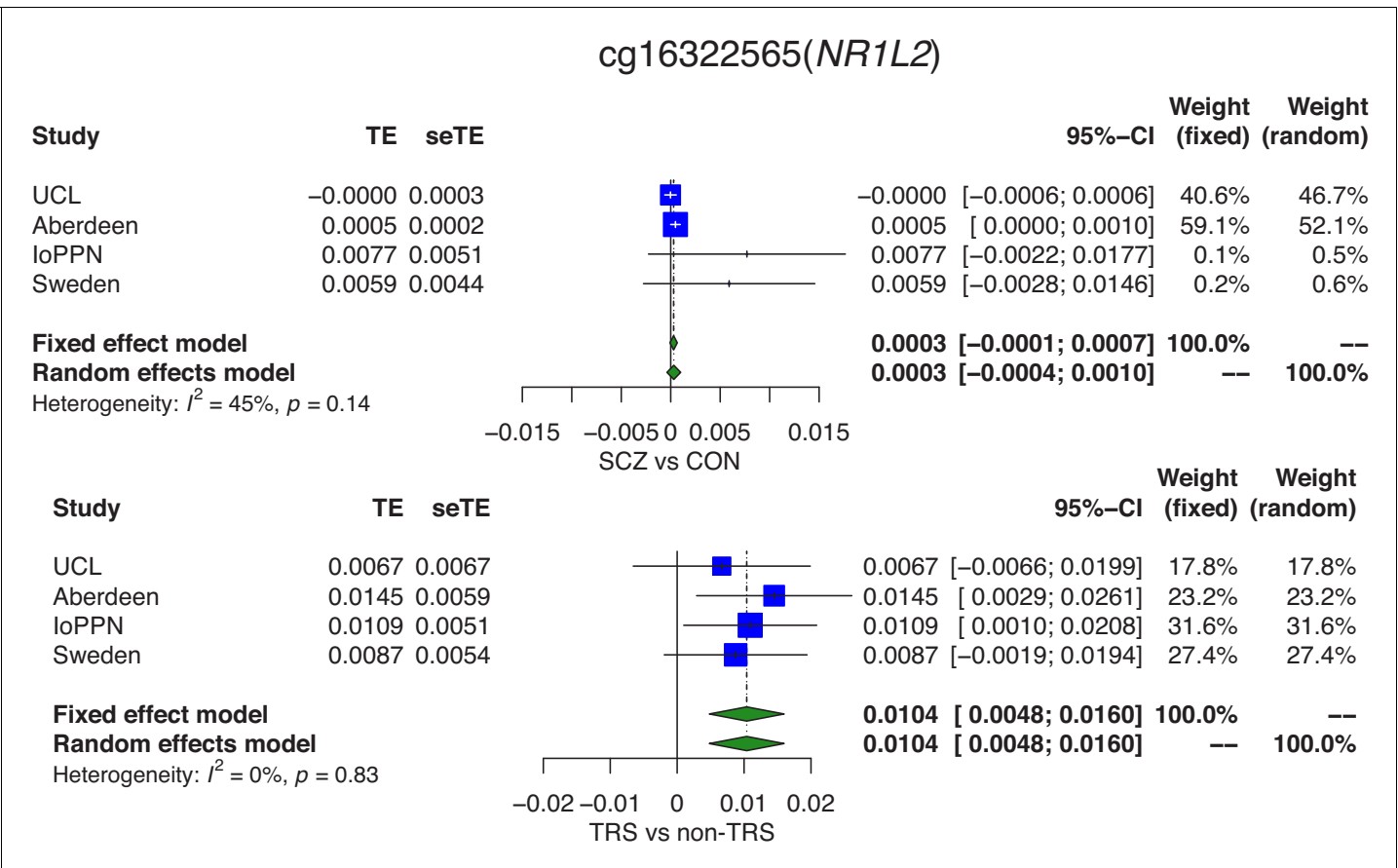

**Figure 8.** Differences in DNA methylation between schizophrenia cases and controls are partially influenced by a subset of cases with treatment-resistant schizophrenia (TRS). Forest plots from a meta-analysis of differences in DNA methylation at cg16322565 located in the *NR1L2* gene on chromosome 3 between (**A**) schizophrenia patients and controls and (**B**) TRS patients prescribed clozapine and non-TRS prescribed other medications. TE: treatment effect (i.e., the mean difference between cases and controls); seTE: standard error of the treatment effect.

The online version of this article includes the following figure supplement(s) for figure 8:

**Figure supplement 1.** Forest plot of a site where DNA methylation is significantly associated with schizophrenia and within cases, with treatment-resistant schizophrenia.

**Figure supplement 2.** Forest plot of a site where DNA methylation is significantly associated with schizophrenia and within cases, with treatment-resistant schizophrenia but with an opposite directions of effect.

reported a decrease in the proportion of natural killer cells and increase in the proportion of granulocytes in schizophrenia patients, with the large number of samples enabling us to identify additional associations with other cell types. We also confirm findings from an independent study of schizophrenia which reported significantly increased proportions of granulocytes and monocytes, and decreased proportions of CD8[+] T-cells using estimates derived from DNA methylation data (*Montano et al., 2016*). Of note, because we can only derive proportion of cell types from whole blood DNA methylation data, and not actual counts, an increase in one or more cell types must be balanced by a decrease in one or more other cell types and an apparent change in the proportion of one specific cell type does not mean that the actual abundance of that cell type is altered. Despite this, the results from DNA methylation-derived cell proportions are consistent with previous studies based on empirical cell abundance measures which have reported increased monocyte counts (*Beumer et al., 2012*; *Moody and Miller, 2018*), increased neutrophil counts (*Garcia-Rizo et al., 2019*; *Núñez et al., 2019*), increased monocyte-to-lymphocyte ratio (*Mazza et al., 2020*; *Steiner et al., 2019*), and increased neutrophil-to-lymphocyte ratio (*Karageorgiou et al., 2019*; *Mazza et al., 2020*) in both schizophrenia and FEP patients compared to controls. Previous studies have also shown that higher neutrophil counts in schizophrenia patients correlate with a greater

burden of positive symptoms (*Núñez et al., 2019*) suggesting that variations in the number of neutrophils is a potential marker of disease severity (*Steiner et al., 2019*). Our sub-analysis of TRS, which is associated with a higher number of positive symptoms (*Bachmann et al., 2017*), found that the increase in granulocytes was primarily driven by those with the more severe phenotype, supporting this hypothesis. Importantly, the differences we observe may actually reflect the effects of various antipsychotic medications that have been previously shown to influence cell proportions in blood (*Steiner et al., 2019*) or a recruitment bias whereby patients with low levels of granulocytes are not prescribed clozapine given the risk of agranulocytosis.

We also identified a highly significant increase in a DNA methylation-derived smoking score in patients with schizophrenia, replicating our previous finding (*Hannon et al., 2016a*). The smoking score captures multiple aspects of tobacco smoking behavior including both current smoking status and the quantity of cigarettes smoked; our results therefore reflect existing epidemiological evidence demonstrating that schizophrenia patients not only smoke more, but also smoke more heavily (*de Leon et al., 2002*; *de Leon and Diaz, 2005*; *McClave et al., 2010*). We also report an increased smoking score in patients with FEP, although not to the same extent as seen in schizophrenia, consistent with a meta-analysis reporting high levels of smoking in FEP (*Myles et al., 2012*). In the subset of treatment-resistant patients, we found that there was an additional increase in smoking score relative to schizophrenia cases prescribed alternative medications, supporting evidence for higher rates of smoking in TRS groups relative to treatment-responsive schizophrenia patients (*Kennedy et al., 2014*). These results not only highlight physiological (i.e., cell proportions) and environmental (i.e., smoking) differences associated with psychosis and schizophrenia and the utility of DNA methylation data for deriving these variables in epidemiological studies, but also highlight the importance of controlling for these differences as potential confounders in analyses of disease-associated DNA methylation differences.

Our EWAS, building on our previous analysis on a subset of the sample cohorts profiled here (*Hannon et al., 2016a*), identified 95 DMPs associated with psychosis that are robust to differences in measured smoking exposure and heterogeneity in blood cellular composition derived from DNA methylation data. Of note, we identified a dramatic increase in sites characterized by an increase in DNA methylation in patients. A key strength of our study is the inclusion of the full spectrum of schizophrenia diagnoses, from FEP through to treatment-resistant cases prescribed clozapine. While this may introduce heterogeneity into our primary analyses, we used a random effects meta-analysis to identify consistent effects across all cohorts and diagnostic subtypes. We also performed an additional analysis focused specifically on cases with a more refined diagnosis of schizophrenia excluding those with FEP, which identified over 1000 DMPs. A number of the top-ranked DMPs are annotated to genes that have a direct relevance to the etiology of schizophrenia, and GO analysis highlighted multiple pathways previously implicated in schizophrenia, including several related to the extracellular matrix (*Berretta, 2012*) and cell-cell adhesion (*O'Dushlaine et al., 2011*). Given the known genetic component to the etiology of schizophrenia, it is interesting that schizophrenia-associated DMPs were found to colocalize to several regions nominated by genetic association studies. Our results suggest that this analysis of a more specific phenotype in a smaller number of samples is potentially more powerful and that schizophrenia cases have a more discrete molecular phenotype that might reflect both etiological factors but also factors associated with a diagnosis of schizophrenia (e.g., medications, stress, etc.). The mean difference in DNA methylation between cases and controls for both psychosis and schizophrenia was small, consistent with other blood-based EWAS of schizophrenia (*Montano et al., 2016*) and complex traits (*Hannon et al., 2018c*; *Hannon et al., 2019b*; *Marioni et al., 2018*) in general. While individually they may be too small to have a strong predictive power as a biomarker, together they may have utility as a molecular classifier (*Chen et al., 2020*).

To explore whether schizophrenia-associated differences in DNA methylation are present before a formal diagnosis of schizophrenia, we also performed an EWAS of individuals with FEP. Strikingly, the majority of our schizophrenia-associated DMPs were found to have a consistent direction of effect in the EWAS of individuals with FEP. While this result is consistent with schizophrenia-associated differences being present prior to a formal diagnosis of schizophrenia, it is not sufficient to state that they are causal; they may still reflect some underlying environmental risk factors or be a consequence of having FEP (e.g., medication exposure or other psychiatric condition). Further work

is needed to explore the extent to which the DMPs associated with psychosis and schizophrenia in this meta-analysis might have a causal role in disease.

Finally, we also report the first systematic analysis of individuals with TRS, identifying seven DMPs at which differential DNA methylation was significantly different in the subset of schizophrenia cases prescribed clozapine. These data are informative for the interpretation of our schizophrenia-associated differences, because a number of these DMPs are driven by the subset of patients on clozapine. Furthermore, a number of sites show opposite effects in our analyses of TRS vs. our analysis of schizophrenia, suggesting they might represent important differences between diagnostic groups. Because the prescription of clozapine is generally only undertaken in patients with TRS, we are unable to separate the effects of clozapine exposure from differences associated with a more severe sub-type of schizophrenia such as the influence of polypharmaceutical treatment.

Our results should be considered in light of a number of important limitations. First, our analyses were constrained by the technical limitations of the Illumina 450K and EPIC arrays, which assay only ~ 3% of CpG sites in the genome. Second, this is a cross-sectional study and it was not possible to distinguish cause from effect. It is possible, and indeed likely, for example, that the differences associated with both schizophrenia and TRS reflect the effects of medication exposure or other consequences of having schizophrenia, for example, living more stressful lives, poorer diet, and health. The importance of such confounding variables is demonstrated by our findings of differential smoking score and blood cell proportions derived directly from the DNA methylation data, although these examples also highlight the potential utility of such effects for molecular epidemiology. Third, although our aim was not to make inferences about mechanistic changes in the brain associated with psychosis, it is important to note that our study analyzed DNA methylation profiled in peripheral blood and therefore can provide only limited information about variation in the primary tissue associated with disease (*Hannon et al., 2015*). Although this limits mechanistic conclusions about the role of DNA methylation in schizophrenia, biomarkers, by definition, need to be measured in an easily accessible tissue and do not need to reflect the underlying pathogenic process. Furthermore, because most classifiers used to quantify variables such as smoking exposure and age have been trained in blood, this represents the optimal tissue in which to derive these measures. Of course, blood may also be an appropriate choice for investigating medication effects, particularly given the known effects on white blood cell counts associated with taking clozapine (*Alvir et al., 1993*). Fourth, while we have explored the potential effects of clozapine on DNA methylation by assessing a sub-group of individuals with TRS, this is just one of a range of antipsychotics schizophrenia and psychosis patients are prescribed. The fact that the TRS group show more extreme differences for many of the schizophrenia-associated DMPs suggests that the polypharmaceutical treatment regimens often prescribed to schizophrenia patients may produce specific DNA methylation signatures in patients, akin to the effect seen for smoking. Fifth, although we found no evidence for a significant interaction between sex and DNA methylation at DMPs associated with schizophrenia, it is possible that there are other DNA methylation differences associated with disease only in males or females. Finally, although we found some evidence that schizophrenia-associated DMPs colocalize to regions nominated by GWAS, the integration of our DNA methylation data with genetic data was beyond the scope of this analysis. Of note, we have previously used mQTL associations to identify discrete sites of regulatory variation associated with schizophrenia risk variants to prioritize specific genes within broad GWAS regions (*Hannon et al., 2016a*; *Hannon et al., 2018a*; *Hannon et al., 2016b*; *Hannon et al., 2017*), and future work will aim to further explore interactions between genetic and epigenetic risk factors.

In conclusion, our analysis of 4483 participants represents the largest study of blood-based DNA methylation in schizophrenia and psychosis yet performed, and one of the largest EWAS studies for *any* disease phenotype. Our study also includes the first within-case analysis of TRS yet performed, providing important molecular insights into genomic differences associated with poor outcome to standard therapeutic approaches. Our results highlight differences in measures of blood cellular composition and smoking behavior derived from methylomic data between not just cases and controls, but also between TRS patients prescribed clozapine and those prescribed alternative medications. We report widespread differences in DNA methylation in psychosis and schizophrenia, a subset of which are driven by the more severe treatment-resistant subset of patients. On a practical level, our study not only demonstrates the utility of DNA methylation data for deriving measures of specific physiological phenotypes (e.g., blood cell-type proportions) and environmental exposures

(e.g., exposure to tobacco smoke) that can be used to identify epidemiological associations with health and disease, but also highlights the importance of properly controlling for these potential confounders in EWAS analyses. Our results are important because they suggest there are also clear molecular signatures of schizophrenia and psychosis that can be identified in whole blood DNA. Although it is unlikely these differences are mechanistically related to neuropathological changes in the brain, they may have utility as diagnostic and prognostic biomarkers in individuals with FEP and may potentially be used to differentiate individuals with TRS at an early stage of disease. Future work should aim to prospectively profile DNA methylation in individuals at risk for FEP and schizophrenia to explore how methylomic variation at baseline can predict outcome and the extent to which longitudinal changes at psychosis-associated DMPs map on to clinical trajectories.

## Materials and methods

### Cohort descriptions

#### University College London samples

Four hundred and forty-seven schizophrenia cases and 456 controls from the University College London schizophrenia sample cohort were selected for DNA methylation profiling. A full description of this cohort can be found elsewhere (*Datta et al., 2010*) but briefly comprises unrelated ancestrally matched cases and controls from the United Kingdom. Case participants were recruited from UK NHS mental health services with a clinical ICD-10 diagnosis of schizophrenia. All case participants were interviewed with the Schedule for Affective Disorders and Schizophrenia-Lifetime Version (SADS-L) (*Spitzer and Endicott, 1977*) to confirm Research Diagnostic Criteria (RDC) diagnosis. A control sample screened for an absence of mental health problems was recruited. Each control subject was interviewed to confirm that they did not have a personal history of an RDC-defined mental disorder or a family history of schizophrenia, bipolar disorder, or alcohol dependence. UK National Health Service multicenter and local research ethics approval was obtained and all subjects signed an approved consent form after reading an information sheet.

#### Aberdeen samples

Four hundred and eighty-two schizophrenia cases and 468 controls from the Aberdeen schizophrenia sample were selected for DNA methylation profiling. The Aberdeen case-control sample has been fully described elsewhere (*International Schizophrenia Consortium, 2008*) but briefly contains schizophrenia cases and controls who have self-identified as born in the British Isles (95% in Scotland). All cases met the Diagnostic and Statistical Manual for Mental Disorders-IV edition (DSM-IV) and International Classification of Diseases 10th edition (ICD-10) criteria for schizophrenia. Diagnosis was made by Operational Criteria Checklist (OPCRIT). Controls were volunteers recruited through general practices in Scotland. Practice lists were screened for potentially suitable volunteers by age and sex and by exclusion of subjects with major mental illness or use of neuroleptic medication. Volunteers who replied to a written invitation were interviewed using a short questionnaire to exclude major mental illness in individual themselves and first-degree relatives. All cases and controls gave informed consent. The study was approved by both local and multiregional academic ethical committees.

#### Monozygotic twins discordant for schizophrenia

The Twins cohort is a multicenter collaborative project aimed at identifying DNA methylation differences in monozygotic twin pairs discordant for a diagnosis of schizophrenia. Ninety-six informative twin pairs (n = 192 individuals) were identified from European twin studies based in Utrecht (The Netherlands), Helsinki (Finland), London (United Kingdom), Stockholm (Sweden), and Jena (Germany). Of the monozygotic twin pairs utilized in the analysis, 75 were discordant for diagnosed schizophrenia, 6 were concordant for schizophrenia, and 15 twin pairs were free of any psychiatric disease. Each twin study has been approved; ethical permission was given by the relevant local ethics committee and the participating twins have provided written informed consent.

## Dublin samples

Three hundred and sixty-one schizophrenia cases and 346 controls were selected from the Irish Schizophrenia Genomics consortium; a detailed description of this cohort can be found in *Morris et al., 2014*. Briefly, participants from the Republic of Ireland or Northern Ireland were interviewed using a structured clinical interview, and diagnosis of schizophrenia or a related disorder (schizoaffective disorder; schizophreniform disorder) was made by the consensus lifetime best estimate method using DSM-IV criteria. Control subjects were ascertained with written informed consent from the Irish GeneBank and represented blood donors from the Irish Blood Transfusion Service. Ethics Committee approval for the study was obtained from all participating hospitals and centers.

## IoPPN samples

The IoPPN cohort comprises 290 schizophrenia cases, 308 FEP patients, and 203 non-psychiatric controls recruited from the same geographical area into three studies via the South London and Maudsley Mental Health National Health Service (NHS) Foundation Trust. Established schizophrenia cases were recruited to the Improving Physical Health and Reducing Substance Use in Severe Mental Illness (IMPACT) study from three English mental health NHS services (*Gaughran et al., 2019*). FEP patients were recruited to the GAP study (*Di Forti et al., 2015*) via in-patient and early intervention in psychosis community mental health teams. All patients aged 18–65 years who presented with an FEP to the Lambeth, Southwark, and Croydon adult in-patient units of the South London and Maudsley Mental Health NHS Foundation Trust between May 1, 2005, and May 31, 2011, who met ICD-10 criteria for a diagnosis of psychosis (codes F20–F29 and F30–F33). Clinical diagnosis was validated by administering the Schedules for Clinical Assessment in Neuropsychiatry (SCAN). Cases with a diagnosis of organic psychosis were excluded. Healthy controls were recruited into the GAP study from the local population living in the area served by the South London and Maudsley Mental Health NHS Foundation Trust, by means of internet and newspaper advertisements, and distribution of leaflets at train stations, shops, and job centers. Those who agreed to participate were administered the Psychosis Screening Questionnaire (*Bebbington and Nayani, 1995*) and excluded if they met criteria for a psychotic disorder or reported to have received a previous diagnosis of psychotic illness. All participants were included in the study only after giving written informed consent. The study protocol and ethical permission was granted by the Joint South London and Maudsley and the Institute of Psychiatry NHS Research Ethics Committee (17/NI/0011).

## Sweden samples

One hundred and ninety schizophrenia cases and 190 controls from the Sweden Schizophrenia Study (S3) were selected for DNA methylation profiling, as described previously (*Kowalec et al., 2019*). Briefly, S3 is a population-based cohort of individuals born in Sweden including 4936 schizophrenia cases and 6321 healthy controls recruited between 2004 and 2010. Schizophrenia cases were identified from the Sweden Hospital Discharge Register with $\geq 2$ hospitalizations with an ICD discharge diagnosis of schizophrenia or schizoaffective disorder . Controls were also selected through Swedish Registers and were group-matched by age, sex, and county of residence and had no lifetime diagnoses of schizophrenia, schizoaffective disorder, or bipolar disorder or antipsychotic prescriptions. Blood samples were drawn at enrollment. All subjects were 18 years of age or older and provided written informed consent. Ethical permission was obtained from the Karolinska Institutet Ethical Review Committee in Stockholm, Sweden.

## The European Network of National Schizophrenia Networks Studying Gene-Environment Interactions cohort (EU-GEI)

Four hundred and fifty-eight FEP cases and 558 controls from the incidence and case-control work package (WP2) of the European Network of National Schizophrenia Networks Studying Gene-Environment Interactions (EU-GEI) cohort were selected for DNA methylation profiling (*Jongsma et al., 2018*). Patients presenting with FEP were identified, between May 1, 2010, and April 1, 2015, by trained researchers who carried out regular checks across the 17 catchment area Mental Health Services across six European countries. FEPs were included if (1) age 18–64 years and (2) resident within the study catchment areas at the time of their first presentation and with a diagnosis of psychosis

(ICD-10: F20–33). Using the Operational Criteria Checklist algorithm (*McGuffin et al., 1991*; *Quattrone et al., 2019*), all cases interviewed received a research-based diagnosis. FEPs were excluded if (1) previously treated for psychosis, (2) they met criteria for organic psychosis (ICD-10: F09), or for a diagnosis of transient psychotic symptoms resulting from acute intoxication (ICD-10: F1X.5). FEPs were approached via their clinical team and invited to participate in the assessment. Random and Quota sampling strategies were adopted to guide the recruitment of controls from each of the sites. The most accurate local demographic data available were used to set quotas for controls to ensure the samples' representativeness of each catchment area's population at risk. Controls were excluded if they had received a diagnosis of, and/or treatment for, a psychotic disorder. All participants provided written informed consent. Ethical approval was provided by relevant research ethics committees in each of the study sites.

## Genome-wide quantification of DNA methylation

Approximately 500 ng of blood-derived DNA from each sample was treated with sodium bisulfite in duplicate, using the EZ-96 DNA methylation kit (Zymo Research, CA, USA). DNA methylation was quantified using either the Illumina Infinium HumanMethylation450 BeadChip (Illumina Inc, CA, USA) or Illumina Infinium HumanMethylationEPIC BeadChip (Illumina Inc) run on an Illumina iScan System (Illumina) using the manufacturers' standard protocol. Samples were batched by cohort and randomly assigned to chips and plates to ensure equal distribution of cases and controls across arrays and minimize batch effects. For the Twins cohort, both members of the same twin pair were run on the same chip. A fully methylated control sample (CpG Methylated HeLa Genomic DNA; New England BioLabs, MA, USA) was included in a random position on each plate to facilitate plate tracking. Signal intensities were imported in R programming environment using the *methylumIDAT* function in the *methylumi* package (*Davis et al., 2015*). Our stringent quality control pipeline included the following steps: (1) checking methylated and unmethylated signal intensities, excluding samples where this was <2500; (2) using the control probes to ensure the sodium bisulfite conversion was successful, excluding any samples with median <90; (3) identifying the fully methylated control sample was in the correct location; (4) all tissues predicted as of blood origin using the tissue prediction from the Epigenetic Clock software (https://DNAmAge.genetics.ucla.edu/) (*Horvath, 2013*); (5) multidimensional scaling of sites on X and Y chromosomes separately to confirm reported gender; (6) comparison with genotype data across SNP probes; (7) *pfilter* function from wateRmelon package (*Pidsley et al., 2013*) to exclude samples with >1% of probes with detection p-value>0.05 and probes with >1% of samples with detection p-value>0.05. PCs were used (calculated across all probes) to identify outliers, samples >2 standard deviations from the mean for both PC1 and PC2 were removed. An additional QC step was performed in the Twins cohort using the 65 SNP probes to confirm that twins were genetically identical. Normalization of the DNA methylation data was performed using the *dasen* function in the *wateRmelon* package (*Pidsley et al., 2013*). As cell count data were not available for these DNA samples, these were estimated from the 450K DNA methylation data using both the Epigenetic Clock software (*Horvath, 2013*) and Houseman algorithm (*Houseman et al., 2012*; *Koestler et al., 2013*), including the seven variables recommended in the documentation for the Epigenetic Clock in the regression analysis. For cohorts with the EPIC array DNA methylation data, we were only able to generate the six cellular composition variables using the Houseman algorithm (*Houseman et al., 2012*; *Koestler et al., 2013*), which were included as covariates. Similarly as smoking data was incomplete for the majority of cohorts, we calculated a smoking score from the data using the method described by *Elliott et al., 2014* and successfully used in our previous (phase 1) analyses (*Hannon et al., 2016a*). Raw and processed data for the UCL, Aberdeen, Dublin, IoPPN, and EU-GEI cohorts are available through GEO accession numbers GSE84727, GSE80417, GSE147221, GSE152027, and GSE152026, respectively.

## Data analysis

All analyses were performed with the statistical language R unless otherwise stated. Custom codes for all steps of the analysis are available on GitHub https://github.com/ejh243/SCZEWAS/tree/master/Phase2; *Hannon, 2021*; copy archived at swh:1:rev:006e92b11dbd3eb7e75dcc173853010fa93461a5.

## Comparison of estimates of cellular composition and tobacco smoking derived from DNA methylation data

A linear regression model was used to test for differences in 10 cellular composition variables estimated from the DNA methylation data, reflecting either proportion or abundance of blood cell types. These estimated cellular composition variables were regressed against case/control status with covariates for age, sex, and smoking. Estimated effects and standard errors were combined across the cohorts using a random effects meta-analysis implemented with the meta package (*Schwarzer, 2007*). The same methodology was used to test for differences in the smoking score derived from DNA methylation data between cases and controls including covariates for age and sex. p-Values are from two-sided tests.

## Within-cohort EWAS analysis

A linear regression model was used to test for differentially methylated sites associated with schizophrenia or FEP. DNA methylation values for each probe were regressed against case/control status with covariates for age, sex, derived cellular composition scores (from the DNA methylation data), derived smoking score (from the DNA methylation data), and experimental batch. For the EU-GEI cohort, there was an additional covariate for contributing study. For the Twins cohort, a linear model was used to generate regression coefficients, but p-values were calculated with clustered standard errors using the *plm* package (*Croissant and Millo, 2008*), recognizing individuals from the same twin pair.

## Within-patient EWAS of clozapine prescription

Four individual cohorts (UCL, Aberdeen, IoPPN, and Sweden) had information on medication and/or clozapine exposure and were included in the TRS EWAS. TRS patients were defined as any case that had ever been prescribed clozapine, and non-TRS patients were defined as schizophrenia cases that had no record of being prescribed clozapine. Within each cohort, DNA methylation values for each probe were regressed against TRS status with covariates for age, sex, cell composition, smoking status, and batch as described for the case-control EWAS.

## Multiple regression analysis of schizophrenia and clozapine prescription

Using the four cohorts that were included in the TRS EWAS (UCL, Aberdeen, IoPPN, and Sweden), we fitted a multiple regression model with two binary indicator variables: one that identified the schizophrenia patients and a second that identified the TRS patients. Within each cohort, DNA methylation values for each probe were regressed against these two binary variables, with covariates for age, sex, derived cellular composition scores (from the DNA methylation data), derived smoking score (from the DNA methylation data), and experimental batch as described above for the other EWAS analyses.

## Meta-analysis

The EWAS results from each cohort were processed using the *bacon* R package (*van Iterson et al., 2017*), which uses a Bayesian method to adjust for inflation in EWAS p-values. All probes analyzed in at least two studies were taken forward for meta-analysis. This was performed using the *metagen* function in the R package meta (*Schwarzer, 2007*), using the effect sizes and standard errors adjusted for inflation from each individual cohort to calculate weighted pooled estimates and test for significance. p-Values are from two-sided tests and significant DMPs were identified from a random effects model at a significance threshold of $9 \times 10^{-8}$, which controls for the number of independent tests performed when analysis data generated with the EPIC array (*Mansell et al., 2019*). DNA methylation sites were annotated with location information for genome build hg19 using the Illumina manifest files (CHR and MAPINFO).

## Overlap with schizophrenia GWAS loci

The GWAS regions were taken from the largest published schizophrenia GWAS to date by *Pardiñas et al., 2018* made available through the Psychiatric Genomics Consortium (PGC) website (https://www.med.unc.edu/pgc/results-and-downloads). Briefly, regions were defined by performing a 'clumping' procedure on the GWAS p-values to collapse multiple correlated signals (due to linkage

disequilibrium) surrounding the index SNP (i.e., with the smallest p-value) into a single associated region. To define physically distinct loci, those within 250 kb of each other were subsequently merged to obtain the final set of GWAS regions. The outermost SNPs of each associated region defined the start and stop parameters of the region. Using the set of 158 schizophrenia-associated genomic loci, we used Brown's method (*Brown, 1975*) to calculate a combined p-value across all probes located within each region (based on hg19) using the probe-level p-values and correlation coefficients between all pairs of probes calculated from the DNA methylation values. Briefly, correlation statistics were calculated and (along with the p-values) were inputted into Brown's formula. As correlations between probes could only be calculated using probes profiled on the same array, this analysis was limited to probes included on the EPIC array. Correlations between probes were calculated within the EU-GEI cohort as this had the largest number of samples.

## Enrichment analyses

Enrichment of the heritability statistics of DMPs was performed against a background set of probes selected to match the distribution of the test set for both mean and standard deviation. This was achieved by splitting all probes into 10 equally sized bins based on their mean methylation level and 10 equally sized bins based on their standard deviation, to create a matrix of 100 bins. After counting the number of DMPs within each bin, we selected the same number of probes from each bin for the background comparison set. This was repeated multiple times, without replacement, until all the probes from at least one bin were selected giving the maximum possible number of background probes (n = 42,968) such that they matched the characteristics of the test set of DMPs.

## GO analysis

Illumina UCSC gene annotation, which is derived from the genomic overlap of probes with RefSeq genes or up to 1500 bp of the transcription start site of a gene, was used to create a test gene list from the DMPs for pathway analysis. Where probes were not annotated to any gene (i.e., in the case of intergenic locations), they were omitted from this analysis, and where probes were annotated to multiple genes, all were included. A logistic regression approach was used to test if genes in this list predicted pathway membership, while controlling for the number of probes that passed quality control (i.e., were tested) annotated to each gene. Pathways were downloaded from the GO website (http://geneontology.org/) and mapped to genes including all parent ontology terms. All genes with at least one 450K probe annotated and mapped to at least one GO pathway were considered. Pathways were filtered to those containing between 10 and 2000 genes. After applying this method to all pathways, the list of significant pathways (p<0.05) was refined by grouping to control for the effect of overlapping genes. This was achieved by taking the most significant pathway and retesting all remaining significant pathways while controlling additionally for the best term. If the test genes no longer predicted the pathway, the term was said to be explained by the more significant pathway, and hence these pathways were grouped together. This algorithm was repeated, taking the next most significant term, until all pathways were considered as the most significant or found to be explained by a more significant term.

## Acknowledgements

This work was primarily supported by grants from the UK Medical Research Council (MRC; MR/K013807/1 and MR/R005176/1) to JM. High-performance computing was supported by MRC Clinical Research Infrastructure Funding (MR/M008924/1). The Finnish Twin study was supported by the Academy of Finland Centre of Excellence in Complex Disease Genetics (grant numbers 213506, 129680), and JK by the Academy of Finland grants 265240, 263278, and 312073. Financial support for the Sweden twin study was provided by the Karolinska Institutet (ALF 20090183 and ALF 20100305 to Hultman) and NIH (R01 MH52857). Collection of the Sweden case-control samples was supported by the Sweden Research Council (Vetenskapsrådet, award D0886501 to PFS) and the NIMH (R01MH077139). Collection of the Irish case-control samples was funded by the Wellcome Trust Case Control Consortium 2 project (085475/B/08/Z and 085475/Z/08/Z), the Wellcome Trust (072894/Z/03/Z, 090532/Z/09/Z, and 075491/Z/04/B), and Science Foundation Ireland (08/IN.1/B1916). The European Network of National Schizophrenia Networks Studying Gene-Environment Interactions (EU-GEI) Project is funded by grant agreement HEALTH-F2-2010-241909 (Project EU-

GEI) from the European Community's Seventh Framework Programme. The IMPaCT programme at King's College London and the South London and Maudsley NHS Foundation Trust is funded by the National Institute for Health Research (RP-PG-0606–1049). The CRESTAR project received funding from the European Union's Seventh Framework Programme for research, technological development, and demonstration under grant agreement 279227 (CRESTAR Consortium). Cardiff University researchers were supported by Medical Research Council (MRC) Centre (G0800509) and Programme Grant (G0801418). Bart PF Rutten is supported by a VIDI grant (number 91718336) from the Netherlands Organisation for Scientific Research. FG is in part supported by the National Institute for Health Research's (NIHR) Biomedical Research Centre at South London and Maudsley NHS Foundation Trust and King's College London, the Stanley Medical Research Institute, the Maudsley Charity, and the National Institute for Health Research (NIHR) Applied Research Collaboration South London (NIHR ARC South London) at King's College Hospital NHS Foundation Trust. MDF and DQ are funded by an MRC fellowship to MDF (MR/M008436/1). We gratefully acknowledge capital equipment funding from the Maudsley Charity (grant no. 980) and Guy's and St Thomas's Charity (grant no. R130505). This study presents independent research supported by the National Institute for Health Research NIHR BioResource Centre Maudsley at South London and Maudsley NHS Foundation Trust and King's College London. The views expressed are those of the author(s) and not necessarily those of the NHS, NIHR, Department of Health and Social Care or King's College London.

## Additional information

### Competing interests

Marta Di Forti: M Di Forti reports personal fees from Janssen, outside the submitted work. Fiona Gaughran: Fiona Gaughran has received honoraria from Lundbeck, Otsuka, and Sunovion. She has a family member with professional links to Lilly and GSK, including shares. Kaarina Kowalec: Kaarina Kowalec has consulted with Emerald Lake Safety Ltd. (2017-2018) and has received speaker honoraria from Biogen/Fraser Health Multiple Sclerosis Clinic (2018). Both are unrelated to the work published here. James MacCabe: James MacCabe has received research funding from H Lundbeck. Michael C O'Donovan: Michael C O'Donovan is supported by a collaborative research grant from Takeda Pharmaceuticals. Takeda played no part in the conception, design, implementation, or interpretation of this study. Patrick Sullivan: PF Sullivan reports the following potentially competing financial interests. Current: Lundbeck (advisory committee, grant recipient). Past three years: Pfizer (scientific advisory board). David Andrew Collier: David A Collier is a full time employee and stockholder of Eli Lilly and Company. Robin M Murray: Robin M Murray reports personal fees from Janssen, Lundbeck, Sunovion, Recordati and Otsuka. The other authors declare that no competing interests exist.

### Funding

| Funder | Grant reference number | Author |
|---|---|---|
| Medical Research Council | MR/K013807/1 | Jonathan Mill |
| Medical Research Council | MR/R005176/1 | Jonathan Mill |
| Medical Research Council | Clinical Research Infrastructure Funding (MR/M008924/1) | Jonathan Mill |
| Karolinska Institutet | ALF 20090183 | Christina M Hultman |
| Karolinska Institutet | ALF 20100305 | Christina M Hultman |
| National Institutes of Health | R01 MH52857 | Christina M Hultman |
| National Institute of Mental Health | R01MH077139 | Patrick Sullivan |
| Finland Centre of Excellence in Complex Disease Genetics | 213506 | Jaakko Kaprio |
| Finland Centre of Excellence in Complex Disease Genetics | 129680 | Jaakko Kaprio |

| Academy of Finland | 265240 | Jaakko Kaprio |
|---|---|---|
| Academy of Finland | 263278 | Jaakko Kaprio |
| Academy of Finland | 312073 | Jaakko Kaprio |
| Vetenskapsrådet | D0886501 | Patrick Sullivan |
| European Union 7th Framework Programme | 279227 | CRESTAR consortium |
| Medical Research Council | Fellowship (MR/M008436/1) | Marta Di Forti |
| NIHR | RP-PG-0606-1049 | Robin M Murray Fiona Gaughran |
| Netherlands Organisation for Scientific Research | VIDI grant (number 91718336) | Bart PF Rutten |

The funders had no role in study design, data collection and interpretation, or the decision to submit the work for publication.

### Author contributions

Eilis Hannon, Data curation, Software, Formal analysis, Investigation, Visualization, Methodology, Writing - original draft, Writing - review and editing; Emma L Dempster, Resources, Data curation, Formal analysis, Validation, Investigation, Methodology, Writing - review and editing; Georgina Mansell, Data curation, Formal analysis, Writing - review and editing; Joe Burrage, Resources, Data curation, Investigation, Writing - review and editing; Nick Bass, Marc M Bohlken, Aiden Corvin, Charles J Curtis, David Dempster, Marta Di Forti, Timothy G Dinan, Gary Donohoe, Fiona Gaughran, Michael Gill, Amy Gillespie, Cerisse Gunasinghe, Hilleke E Hulshoff, Christina M Hultman, Viktoria Johansson, René S Kahn, Jaakko Kaprio, Gunter Kenis, Kaarina Kowalec, James MacCabe, Colm McDonald, Andrew McQuillin, Derek W Morris, Kieran C Murphy, Colette J Mustard, Igor Nenadic, Michael C O'Donovan, Diego Quattrone, Alexander L Richards, Bart PF Rutten, David St Clair, Sebastian Therman, Timothea Toulopoulou, Jim Van Os, John L Waddington, Patrick Sullivan, Evangelos Vassos, Resources, Writing - review and editing; Gerome Breen, David Andrew Collier, Resources, Funding acquisition, Writing - review and editing; Robin M Murray, Conceptualization, Resources, Supervision, Funding acquisition, Methodology, Writing - original draft, Project administration, Writing - review and editing; Leonard S Schalkwyk, Conceptualization, Resources, Funding acquisition, Methodology, Writing - review and editing; Jonathan Mill, Conceptualization, Supervision, Funding acquisition, Methodology, Writing - original draft, Project administration, Writing - review and editing

### Author ORCIDs

Eilis Hannon ⓘ http://orcid.org/0000-0001-6840-072X
Georgina Mansell ⓘ http://orcid.org/0000-0003-2620-1786
Jaakko Kaprio ⓘ https://orcid.org/0000-0002-3716-2455
Andrew McQuillin ⓘ https://orcid.org/0000-0003-1567-2240
Colette J Mustard ⓘ https://orcid.org/0000-0001-5834-2765
Bart PF Rutten ⓘ http://orcid.org/0000-0002-9834-6346
Sebastian Therman ⓘ https://orcid.org/0000-0001-9407-4905
Jonathan Mill ⓘ https://orcid.org/0000-0003-1115-3224

### Decision letter and Author response

Decision letter https://doi.org/10.7554/eLife.58430.sa1
Author response https://doi.org/10.7554/eLife.58430.sa2

## Additional files

### Supplementary files

• Supplementary file 1. Supplementary Tables 1-14.

• Transparent reporting form

## Data availability

Raw and processed data for the UCL, Aberdeen and Dublin cohorts are available through GEO accession numbers GSE84727, GSE80417, and GSE147221, respectively.

The following dataset was generated:

| Author(s) | Year | Dataset title | Dataset URL | Database and Identifier |
|---|---|---|---|---|
| Hannon E, Mill J | 2020 | Blood DNA methylation profiles from schizophrenia cases and controls | https://www.ncbi.nlm.nih.gov/geo/query/acc.cgi?acc=GSE147221 | NCBI Gene Expression Omnibus, GSE147221 |

The following previously published datasets were used:

| Author(s) | Year | Dataset title | Dataset URL | Database and Identifier |
|---|---|---|---|---|
| Hannon E, Dempster E, Viana J, Burrage J, Smith AR, Macdonald R, St. Clair D, Mustard C, Breen G, Therman S, Kaprio J, Toulopoulou T, Hulshoff Pol HE, Bohlken MM, Kahn RS, Nenadic I, Hultman CM, Murray RM, Collier DA, Bass N, Gurling H, McQuillin A, Schalkwyk L, Mill J | 2016 | An integrated genetic-epigenetic analysis of schizophrenia: Evidence for co-localization of genetic associations and differential DNA methylation | https://www.ncbi.nlm.nih.gov/geo/query/acc.cgi?acc=GSE80417 | NCBI Gene Expression Omnibus, GSE80417 |
| Hannon E, Dempster E, Viana J, Burrage J, Smith AR, Macdonald R, St. Clair D, Mustard C, Breen G, Therman S, Kaprio J, Toulopoulou T, Hulshoff Pol HE, Bohlken MM, Kahn RS, Nenadic I, Hultman CM, Murray RM, Collier DA, Bass N, Gurling H, McQuillin A, Schalkwyk L, Mill J | 2016 | An integrated genetic-epigenetic analysis of schizophrenia: Evidence for co-localization of genetic associations and differential DNA methylation | https://www.ncbi.nlm.nih.gov/geo/query/acc.cgi?acc=GSE84727 | NCBI Gene Expression Omnibus, GSE84727 |

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
