## [Decision Letter]

**Acceptance summary:**

This manuscript reports on a cross study analysis of DNA methylation marks in blood samples from over 4,000 individuals suffering from psychosis, schizophrenia and treatment-resistant schizophrenia. Evidence was presented suggesting this approach has future potential as a diagnostic tool to identify physiological and environmental exposure variables contributing to mental illness.

**Decision letter after peer review:**

Thank you for submitting your article "DNA methylation meta-analysis reveals cellular alterations in psychosis and markers of treatment-resistant schizophrenia" for consideration by *eLife*. Your article has been reviewed by three peer reviewers, one of whom is a member of our Board of Reviewing Editors, and the evaluation has been overseen by a Senior Editor.

The reviewers have discussed the reviews with one another and the Reviewing Editor. Together we have concluded that while the analysis seems to be carefully done and this is the largest report of its kind, the work is nonetheless lacking in impact due to the absence of any firm biological or clinical conclusions. The Editors are willing to consider your response to this criticism and request that the conclusions move beyond these associative findings, as well as the changes suggested by reviewers below. Please understand this is not an assurance that your manuscript will be considered acceptable upon revision. We ask you to consider carefully whether your report might be more suitable for a different journal

We would like to draw your attention to changes in our revision policy that we have made in response to COVID-19 (https://elifesciences.org/articles/57162). Because many researchers have temporarily lost access to the labs, we will give authors as much time as they need to submit revised manuscripts.

Summary:

This is the largest study of DNA methylation differences in the blood of controls and patients with psychosis, performed in a sample of 4,483 participants. This is an important piece of work conducted to the highest standards of methodological rigour. By drawing together most case-control DNAm studies of schizophrenia in a single meta-analysis, this work will provide the most up-to-date information for some time, and is likely to generate a lot of interest.

As predictable, the authors found significant differences in measures of blood cell proportions and smoking exposure in patients with psychosis compared with controls, and in patients with schizophrenia with clozapine treatment compared with other patients. They also detected differentially methylated positions in such comparisons. The authors have employed an appropriate methodology to search for schizophrenia- and psychosis- associated methylation changes, and the manuscript is interesting and well-written.

Essential revisions:

A more extensive analysis may increase our insight about DNA methylation differences in schizophrenia, and is therefore necessary.

1) An important question is whether the methylation differences are pre-existing the disorder or a consequence, an epiphenomenon of the disorder. The fact that the authors detect a higher number of DMPs when they exclude individuals with first episode psychosis from their analysis could suggest that the methylation differences are not present before the onset of the disorder. However, the authors have the resources and the ability to better answer this question. For example:

a) I think they should report in a separate section the results in the two samples of FEP individuals compared with age-matched controls. Can they identify any FEP-specific DMP?

b) Also, I think they could try to integrate their data with other blood methylation datasets, to see whether the DMPs associated with psychosis/schizophrenia have been associated with environmental risk factors associated with schizophrenia. For example, the authors could check the overlap of the DMPs with blood methylation changes associated with gestational age (PMID: 32114984; this work contains references to other studies that may be useful too). Data on methylation and cannabis or other environmental factors, if available, may be useful too.

c) The authors could also explore, in patients and controls, the relationship between age and methylation of the DMPs. An increase of the differences between patients and controls in older ages would suggest that the methylation differences are related to factors that are secondary to the disorders, while the presence of methylation differences at younger age could suggest the opposite. Analyzing the interaction between methylation and age on case-control status could be an alternative way to answer this question.

2) Sex is an important biological variable that the authors could analyze more extensively, considering that being male is a risk factor for schizophrenia, and is associated with a different epigenetic regulation. The authors have already the statistics to analyze whether the psychosis/schizophrenia-associated DMPs are also associated with sex. Moreover, they could analyze the interaction between methylation and sex on case-control status and/or perform analyses stratified by sex.

3) The authors did not find association of schizophrenia with age acceleration. However, a recent study has performed a comprehensive analysis of 14 epigenetic clocks categorized according to what they were trained to predict: chronological age, mortality, mitotic divisions, or telomere length. I think it is relevant that the authors try to validate and perhaps extend the findings of Higgis-Chen and coll. ("Schizophrenia and Epigenetic Aging Biomarkers: Increased Mortality, Reduced Cancer Risk, and Unique Clozapine Effects", PMID: 32199607).

4) Adjustment: I have not found any clear information about ethnicity/race. I assume the samples were mainly composed by white Caucasians. Did the authors perform any adjustment for ethnicity/race or population stratification? Also, were principal component of negative control probes included as covariates?

5) Replication: was there any replication at the level of DMP in the data from Montano et al.? Also, if many DMPs are under genetic control, we should expect an overlap between DMPs in blood and brain of patients with schizophrenia. Have the authors analyzed such overlap?

6) I think the authors should be more cautious in interpreting the clozapine data. They write: "Studies have also shown that higher neutrophil counts in schizophrenia patients correlate with a greater burden of positive symptoms (Núñez et al., 2019) suggesting that variations in the number of neutrophils is a potential marker of disease severity(Steiner et al., 2019). Our sub-analysis of treatment-resistant schizophrenia, which is associated with a higher number of positive symptoms (Bachmann et al., 2017), found that the increase in granulocytes was primary driven by those with the more severe phenotype, supporting this hypothesis." Actually, the fact that TRS cases are characterized by a significantly higher proportion of granulocytes could be related a "recruitment bias": because clozapine administration is associated with a risk of agranulocytosis, clozapine is usually not prescribed to patients with low number of granulocytes. I think this possibility needs to be mentioned, unless the authors can exclude it.

7) The Abstract details the (unsurprising) smoking results but lacks other findings, such as the GO analysis and the localisation of findings to previously associated GWAS loci.

8) The authors could consider providing a DNAm-based predictor of SCZ/SCZ-resistance based on their dataset – to be tested in a series of leave-one-out analyses. In my opinion, this would provide further interest in the results, provide evidence of replication somewhat lacking from the current version, and could be used by others to test for SCZ/TRS prediction in their cohorts or for the purpose of PheWAS.

9) There are a large number of findings reported with only a p-value given, and no effect size. In many cases, I think there's no reason that additional info couldn't be added.

10) It's not sufficiently clear in the text how the effects of SCZ were disambiguated from TRS – when the latter group is nested within the first.

11) Whether DNAm is a cause or consequence of liability to SCZ could be further examined in the paper – and I'm not sure why the authors have stopped short of further MR-based tests of this question.

12) The correction for smoking is somewhat heterogeneous across studies (“smoking status”). If they were current non-smokers, was this recent? Further examination of whether reporting findings attenuate after inclusion of AHRR CpGs would provide greater confidence that some are not due to residual confounding. Alcohol and BMI are also likely to give rise to similar issues.

[Editors' note: further revisions were suggested prior to acceptance, as described below.]

Thank you for resubmitting your work entitled "DNA methylation meta-analysis reveals cellular alterations in psychosis and markers of treatment-resistant schizophrenia" for further consideration by *eLife*. Your revised article has been evaluated by Huda Zoghbi (Senior Editor) and a Reviewing Editor.

Summary:

The original submission was recognized as the largest study of DNA methylation differences in the blood of controls and patients with psychosis and was considered an important piece of work but the impact was lessened by many of the observed changes being predictable and due to secondary factors such as smoking or medication. A further weakness was the lack of a clear biological or clinical association with the methylation changes observed.

The manuscript has been improved but there are some remaining issues that need to be addressed before acceptance, as outlined below:

1) The authors are asked to highlight in the Discussion what they believe is the main take-home message of the manuscript. For example, what does this meta-analysis tell us about schizophrenia that we did not know before? or about the potential use of DNA methylation in clinical study? If there are no big answers, what should be the next step forward in studying the epigenetic of schizophrenia? Because this paper is the largest meta-analysis on blood DNA methylation in schizophrenia, these sort of conclusions are expected.

2) Some additional limitations should be mentioned. For example, the interaction of DNA methylation with age and sex was analyzed only for the schizophrenia-associated DMPs (which does not exclude that there could be other methylation differences associated with schizophrenia only in males or females among the remaining positions). Ignoring the possibility that genotype may interact with methylation could be another limitation of current EWASs.

---

## [Author Response]

Essential revisions:A more extensive analysis may increase our insight about DNA methylation differences in schizophrenia, and is therefore necessary.1) An important question is whether the methylation differences are pre-existing the disorder or a consequence, an epiphenomenon of the disorder. The fact that the authors detect a higher number of DMPs when they exclude individuals with first episode psychosis from their analysis could suggest that the methylation differences are not present before the onset of the disorder. However, the authors have the resources and the ability to better answer this question. For example:a) I think they should report in a separate section the results in the two samples of FEP individuals compared with age-matched controls. Can they identify any FEP-specific DMP?

The reviewers make a good point, and we agree that the diverse range of cohorts included in our meta-analysis provides us with the opportunity to more systematically assess whether the DNA methylation (DNAm) differences we observe are present before a diagnosis of schizophrenia (SZ) or whether they are more likely to be a consequence of disease. To address this, we performed an epigenome-wide association study (EWAS) analysis of DNAm differences associated with first episode psychosis (FEP) in the IoPPN and EU-GEI cohorts (total n: 698 FEP cases and 724 controls), meta-analysing the results across 384,217 common DNAm sites. We identified no significant differentially methylated positions (DMPs) after stringently correcting for multiple testing, although this is not surprising given the greatly attenuated sample size (compared to the full schizophrenia meta-analysis), and the fact that the FEP group is likely to be more heterogeneous compared to the diagnosed SZ case group; both these factors negatively influence power to detect effects.

Despite the lack of experiment-wide significant differences amongst these results, they still have potential value for interpreting our SZ-associated DMPs. To facilitate this analysis, we repeated our EWAS of diagnosed SZ, excluding the IoPPN cohort to ensure that there were no overlapping samples between the SZ vs control analysis and the FEP vs control analysis. This analysis identified 125 significant DMPs, of which 101 were also tested in the FEP EWAS. To see if there was any evidence for differential DNAm at these sites prior to a diagnosis of SZ, we compared the estimated differences between SZ cases and controls and FEP cases and controls (Author response image 1). Strikingly, 96 (95.0%) of the tested DMPs had a consistent direction of effect in the FEP EWAS, a significantly higher rate than expected by chance (P = 6.58 x10^-23^). While this observation is consistent with SZ-associated differences being present prior to diagnosis, it is not sufficient to state that they are causal; they may still reflect some underlying environmental risk factor or be a consequence of FEP (e.g. medication exposure). We believe these results are interesting and have added them to the manuscript.

**Author response image 1. sa2fig1:** Comparison of effect sizes from an EWAS of diagnosed SZ (x-axis) and FEP (y-axis) for SZ-associated DMPs. Shown is the mean difference (% DNAm) between cases and controls.

b) Also, I think they could try to integrate their data with other blood methylation datasets, to see whether the DMPs associated with psychosis/schizophrenia have been associated with environmental risk factors associated with schizophrenia. For example, the authors could check the overlap of the DMPs with blood methylation changes associated with gestational age (PMID: 32114984; this work contains references to other studies that may be useful too). Data on methylation and cannabis or other environmental factors, if available, may be useful too.

This was another good suggestion by the Reviewer, and to facilitate this analysis we downloaded the complete database of results from the online EWAS catalog (http://ewascatalog.org/), which incorporates associations between variable DNAm and traits/exposures reported in the literature. Across studies undertaken using blood DNA (isolated from whole blood or cord blood) there were 101,091 significant DMPs (at P < 10^-7^) associated with 87 traits. Of the 1,048 specific DMPs associated with SZ in our meta-analysis, 219 (20.9%) were present in the database and significantly associated with 18 different traits (Author response table 1). Where effect sizes were available, we counted the number of DMPs for which the direction of effect was consistent and used a binomial test to see if there was a significant bias. Interestingly, we found that age associated DMPs overlapping with SZ associated DMPs were enriched for sites with a discordant direction of effect, as were those associated with HDL (Figure 6). In contrast, SZ DMPs also associated with CRP and gestational age had a higher rate of concordance than expected. If SZ-associated differences at these sites reflect exposure to these factors then we might anticipate a dose dependent relationship and a notable correlation in the magnitude of effect. For CRP and HDL we do indeed see a strong correlation between the magnitudes of effect, with weaker correlations for gestational age and age (Figure 6). Likewise, of the 95 DMPs associated with psychosis in our meta-analysis, 39 (41.1%) were present in the database and significantly associated with 11 traits (Author response table 2). Unlike for SZ, however, none of these DMPs shared with these traits were significantly enriched for a concordant/discordant direction of effect. These results for the schizophrenia overlaps have been added to the revised manuscript.

**Author response table 1. resptable1:** Summary of the overlap between SZ-associated DMPs and DMPs associated with other traits in either whole blood or cord blood using results from the online EWAS catalog (http://ewascatalog.org/). Shown are traits with more than one overlapping DMP.

*Trait*	*Total Overlap*	*Discordant direction of effect*	*Concordant direction of effect*	*P-value*
*Age*	*30*	*28*	*2*	*8.68E-07*
*Alcohol consumption per day*	*8*	*7*	*1*	*0.070313*
*Body mass index*	*11*	*0*	*8*	*0.007813*
*C-reactive protein*	*10*	*0*	*10*	*0.001953*
*Gestational age*	*105*	*33*	*72*	*0.000178*
*HIV infection*	*8*	*4*	*4*	*1*
*Inflammatory bowel disease*	*2*	*0*	*0*	*NA*
*Maternal smoking in pregnancy*	*7*	*2*	*5*	*0.453125*
*Primary Sjogrens syndrome*	*31*	*0*	*0*	*NA*
*Rheumatoid arthritis*	*20*	*14*	*6*	*0.115318*
*Serum high-density lipoprotein cholesterol*	*12*	*12*	*0*	*0.000488*
*Serum low-density lipoprotein cholesterol*	*2*	*0*	*2*	*0.5*
*Serum total cholesterol*	*5*	*0*	*5*	*0.0625*
*Sex*	*54*	*0*	*1*	*1*
*Smoking*	*16*	*4*	*1*	*0.375*

**Author response table 2. resptable2:** Summary of the overlap between psychosis-associated DMPs and DMPs associated with other traits in either whole blood or cord blood using results from the online EWAS catalog (http://ewascatalog.org/). Shown are traits with more than one overlapping DMP

*Trait*	*Total Overlap*	*Discordant direction of effect*	*Concordant direction of effect*	*P-value*
*Age*	*3*	*3*	*0*	*0.25*
*Body mass index*	*4*	*3*	*0*	*0.25*
*C-reactive protein*	*5*	*1*	*4*	*0.375*
*Gestational age*	*20*	*9*	*11*	*0.823803*
*HIV infection*	*3*	*2*	*1*	*1*
*Primary Sjogrens syndrome*	*3*	*0*	*0*	*NA*
*Rheumatoid arthritis*	*4*	*0*	*4*	*0.125*
*Sex*	*19*	*0*	*1*	*1*
*Smoking*	*6*	*3*	*0*	*0.25*

c) The authors could also explore, in patients and controls, the relationship between age and methylation of the DMPs. An increase of the differences between patients and controls in older ages would suggest that the methylation differences are related to factors that are secondary to the disorders, while the presence of methylation differences at younger age could suggest the opposite. Analyzing the interaction between methylation and age on case-control status could be an alternative way to answer this question.

The reviewers make an interesting suggestion, and certainly the interaction of age with the estimated effects from our meta-analysis may indeed tell us something about where on the causal pathway the differences we report lie. As requested by the reviewers we have investigated whether there is any evidence of age specific SZ-associated differences in DNA methylation. To do this we took the 1,048 SZ-associated DMPs and refitted our analysis model using an additional interaction term between age and schizophrenia status. This analysis was done individually for each cohort prior to the interaction effects being meta-analysed using a random effects model as described in our original manuscript. Overall, we found limited evidence for a relationship between age and DNAm at SZ-associated DMPs; controlling for multiple testing (P < 0.00004771), only two (0.002%) DNAm sites were identified as showing a significant interaction with age (Author response image 2). However, given the differences in ages between cases and controls in some cohorts, it is plausible that our dataset is not optimal for addressing this question. Although we think that the lack of significant effect here should not be interpreted as an absence of an effect, these results have now been added to the revised manuscript.

**Author response image 2. sa2fig2:** Limited evidence for a relationship between age and DNAm at SZ-associated DMPs. Shown is a histogram of –log10(p-values) for the interaction term between age and schizophrenia status across all 1,048 SZ-associated DMPs.

2) Sex is an important biological variable that the authors could analyze more extensively, considering that being male is a risk factor for schizophrenia, and is associated with a different epigenetic regulation. The authors have already the statistics to analyze whether the psychosis/schizophrenia-associated DMPs are also associated with sex. Moreover, they could analyze the interaction between methylation and sex on case-control status and/or perform analyses stratified by sex.

This is another interesting suggestion. As requested by the reviewers we have investigated whether there is any influence of sex on DNAm across the set of SZ-associated DMPs. To do this we took the 1,048 sites associated with SZ and refitted our analysis model with an additional interaction term between sex and schizophrenia status. This analysis was done within each cohort prior to the interaction effects being meta-analysed using a random effects model as described in our original manuscript. We found no evidence that there was any influence of sex on DNAm at these sites, and none of the sites were associated with a significant interaction (P < 0.00004771) (Author response image 3).

**Author response image 3. sa2fig3:** Limited evidence for a relationship between sex and DNAm at SZ-associated DMPs. Shown is a histogram of –log10(p-values) for the interaction term between sex and schizophrenia status across all 1,048 SZ-associated DMPs.

3) The authors did not find association of schizophrenia with age acceleration. However, a recent study has performed a comprehensive analysis of 14 epigenetic clocks categorized according to what they were trained to predict: chronological age, mortality, mitotic divisions, or telomere length. I think it is relevant that the authors try to validate and perhaps extend the findings of Higgis-Chen and coll. ("Schizophrenia and Epigenetic Aging Biomarkers: Increased Mortality, Reduced Cancer Risk, and Unique Clozapine Effects", PMID: 32199607).

Several of the datasets included in our meta-analysis have already been studied in detail with regard to elevated epigenetic aging (e.g. Kowalec et al., Translational Psychiatry, PMID: 31164630; Wu wt al, Schizophrenia Bulletin, PMID: 33269797) and a meta-analysis of epigenetic aging using a range of clocks in >1,100 cases and 1,200 controls from this study is currently in review and available as a preprint on bioRxiv (https://www.biorxiv.org/content/10.1101/727859v1). Given that the current manuscript already includes a detailed analysis of DNAmAge and PhenoAge across all cohorts – the largest single analysis of accelerated aging in schizophrenia to date – and finds no evidence for any accelerated epigenetic aging, we believe that an analysis across 14 additional epigenetic clocks would be beyond the scope of the current manuscript. Our data is freely-available for anyone who is interested in testing the large (and expanding) number of additional clock algorithms available in the literature. It is worth noting that there are some important limitations to the use of epigenetic clocks to assess accelerated biological aging, especially in older samples (for example, see El Khoury et al., Genome Biology, PMID: 31847916).

4) Adjustment: I have not found any clear information about ethnicity/race. I assume the samples were mainly composed by white Caucasians. Did the authors perform any adjustment for ethnicity/race or population stratification? Also, were principal component of negative control probes included as covariates?

We apologize if we did not provide enough detail about the ethnicity/race of samples included in the study. With the exception of the IoPPN and EU-GEI cohorts, which had some ethnic heterogeneity, the reviewer is correct that the samples profiled in this study were predominantly Caucasian. To illustrate this, we have now used the genetic data available for each sample to explore ethnicity using genetic principal components (PCs) incorporating data from the HapMap project (Figure 1—figure supplement 2). To explore whether ethnicity influences our results, we reanalyzed data from individual cohorts including increasing numbers of genetic PCs to our model. Even in the most ethnically diverse cohort (IoPPN) the inclusion of genetic PCs made very limited difference to the results and there was a very strong correlation between models (Figure 4—figure supplement 1). These details have been included in the revised manuscript.

5) Replication: was there any replication at the level of DMP in the data from Montano et al.? Also, if many DMPs are under genetic control, we should expect an overlap between DMPs in blood and brain of patients with schizophrenia. Have the authors analyzed such overlap?

It was a great suggestion to explore overlap between our findings and those from Montano et al., and we have undertaken additional analyses to explore this. Of note, the raw data and full results tables were not available for this study, so we were somewhat limited in the scope of what was possible. First, our results for cell composition differences are consistent with those reported in Montano et al. (i.e. higher proportions of monocytes and granulocytes and lower proportions of CD8 T-cells) although we also report additional differences in CD4 T-cells and natural killer cells. Second, two of our SZ-associated DMPs (cg00390724 and cg09868768) overlapped with the 172 replicated DMPs reported by Montano et al., with the same direction of effect. As the results for the full dataset were not available, we could not check for overall consistency of direction of effect of our SZ-associated DMPs, although we were able to explore whether our results replicated theirs. Comparing the estimated mean difference between cases and controls for the 172 replicated DMPs reported by Montano et al., we were able to test 167 of these in our data; of note, 119 of these showed the sample direction of effect (Author response image 4), which is a significantly higher rate than expected by chance (P = 3,83x10-8).

**Author response image 4. sa2fig4:** Comparison of results from our EWAS meta-analysis with those for 167 replicated SZ-associated DMPs identified by Montano et al. Overall there is a significant enrichment of consistent results between the analyses for these DMPs. The red crosses highlight the two DNAm sites that were reported as significant by Montano et al. and also passed our multiple testing threshold in our meta-analysis.

We disagree with the hypothesis that because a proportion of our DMPs are under genetic control we would necessarily expect them to overlap between blood and brain. While it is true that genetic effects mediate inter-individual correlations between blood and brain, it does not follow that all genetic effects are conserved between tissues. However, to explore the reviewer’s comment about overlap between blood and brain DMPs, we first leveraged data from our previous study quantifying covariation in DNAm between blood and different brain regions using matched tissue samples (Hannon et al., Epigenetics, PMID: 26457534) to investigate whether our DMPs are likely to be mirrored in brain. Using matched blood and prefrontal cortex (PFC) data we found that of the 1,048 SZ-associated DMPs, no DMPs were characterized by a correlation in DNAm between blood and cortex of > 0.8, only 5 were characterized by a correlation > 0.5, and 100 were characterized by a correlation > 0.2. Second, we performed a direct comparison of our blood based EWAS results with those from a meta-analysis of PFC EWAS datasets (Viana et al., Human Molecular Genetics, PMID: 28011714). Our PFC study is much smaller than our blood meta-analysis and relatively underpowered to detect a large number of true positives; therefore, we explored whether there was any evidence of consistency in our results by comparing the estimated mean differences between schizophrenia cases and controls across both tissues. This analysis showed that 627 of the 1,042 DMPs tested in both analyses had the same direction of effect (Author response image 5), a significantly higher rate than expected by chance (P = 5.43x10-11).

**Author response image 5. sa2fig5:** Comparison of results from our EWAS meta-analysis (x-axis) with those identified in an analysis of human PFC (y-axis). Overall there is a significant enrichment of consistent results between the analyses for SZ-associated DMPs. Black dots represent DMPs that we found to show low covariation (r < 0.2) between blood and cortex tissue in a previous analysis of matched samples from the same individuals (Hannon et al., Epigenetics, PMID: 26457534), with blue (r > 0.2) and red (r > 0.5) symbols highlighting sites with stronger covariation in DNAm between tissues.

6) I think the authors should be more cautious in interpreting the clozapine data. They write: "Studies have also shown that higher neutrophil counts in schizophrenia patients correlate with a greater burden of positive symptoms (Núñez et al., 2019) suggesting that variations in the number of neutrophils is a potential marker of disease severity(Steiner et al., 2019). Our sub-analysis of treatment-resistant schizophrenia, which is associated with a higher number of positive symptoms (Bachmann et al., 2017), found that the increase in granulocytes was primary driven by those with the more severe phenotype, supporting this hypothesis." Actually, the fact that TRS cases are characterized by a significantly higher proportion of granulocytes could be related a "recruitment bias": because clozapine administration is associated with a risk of agranulocytosis, clozapine is usually not prescribed to patients with low number of granulocytes. I think this possibility needs to be mentioned, unless the authors can exclude it.

We agree that our analyses of schizophrenia patients treated with clozapine need to be interpreted carefully and highlight strongly in the discussion that we can’t separate the effects of clozapine exposure from environmental effects or confounders related to the subset of patients that are treatment resistant. However, the reviewer highlights an important bias that we had not considered and we have now added this caveat to the Discussion.

7) The Abstract details the (unsurprising) smoking results but lacks other findings, such as the GO analysis and the localisation of findings to previously associated GWAS loci.

We have added some additional details to the Abstract as suggested.

8) The authors could consider providing a DNAm-based predictor of SCZ/SCZ-resistance based on their dataset – to be tested in a series of leave-one-out analyses. In my opinion, this would provide further interest in the results, provide evidence of replication somewhat lacking from the current version, and could be used by others to test for SCZ/TRS prediction in their cohorts or for the purpose of PheWAS.

We agree that the development of an epigenetic “predictor” for SZ based on our results is interesting – as stated in our Discussion – but believe it to be beyond the scope of the current analysis, especially given the large number of analyses and results already presented in the manuscript. Furthermore, a number of papers on this subject using overlapping data from the current study have been published (e.g., Chen et al., 2020), and a large analysis across multiple datasets has been submitted for publication by our collaborators. Finally, we are wary that there are some important caveats to such an analysis; given the very different characteristics of the cohorts included in our meta-analysis it is not obvious how to train such a predictor and what the results of the model would mean in any given individual cohort.

9) There are a large number of findings reported with only a p-value given, and no effect size. In many cases, I think there's no reason that additional info couldn't be added.

We apologize for this oversight and have now included these in the revised manuscript.

10) It's not sufficiently clear in the text how the effects of SCZ were disambiguated from TRS – when the latter group is nested within the first.

To separate out effects associated with SZ from effects associated with TRS, we fitted a multiple regression model with two binary indicator variables: one that identified the SZ patients and a second that identified the TRS patients. Within each cohort, DNA methylation values for each probe were regressed against these two binary variables, with covariates for age, sex, cell composition, smoking status, and batch as described for the other EWAS. This analysis was only possible for the four cohorts that were included in the TRS EWAS (UCL, Aberdeen, IoPPN and Sweden), and the results from these cohorts were then meta-analysed together, separately for the results for the two binary indicator terms. The inclusion of these two dummy variables allowed us to separately estimate differences associated with all schizophrenia cases and differences only associated with TRS cases. We realise now that we had omitted a description of this analysis from the Materials and methods section of the manuscript; we apologize for this oversight and this has now been rectified in the resubmission.

11) Whether DNAm is a cause or consequence of liability to SCZ could be further examined in the paper – and I'm not sure why the authors have stopped short of further MR-based tests of this question.

Given the dynamic nature of the epigenome, it is clear that the differences we report could be either causal or a consequences of having SZ. The analyses of FEP discussed above are one approach to explore this. While MR has been proposed as a method for testing causal pathways between DNAm and an outcome, it is only possible for sites that are robustly associated with genetic variation. As reported in the manuscript only 256 (24.4%) of the SZ-associated DMPs have an associated mQTL, meaning if we were to apply an MR analysis this would only be possible for a small proportion of our results. Furthermore, the genetic effects at these sites are small, so the power to perform this MR is likely insufficient to draw any meaningful conclusions. In truth we anticipate that many of the associations we report will result from either environmental risk factors for schizophrenia/psychosis or the consequences of having disease (e.g. medication exposure). This is why one of the strengths of our study is the analysis of TRS cases, which enabled us to determine which of the reported DMPs might be driven by medication. We have expanded our discussion about causality in the Discussion section of the manuscript to further address this point.

12) The correction for smoking is somewhat heterogeneous across studies (“smoking status”). If they were current non-smokers, was this recent? Further examination of whether reporting findings attenuate after inclusion of AHRR CpGs would provide greater confidence that some are not due to residual confounding. Alcohol and BMI are also likely to give rise to similar issues.

In fact, the correction for smoking status was consistent across all samples. As described, we used a quantitative smoking score derived from the DNAm data as a covariate in each of the EWAS analyses presented. For samples where self-reported smoking data was also available, there was a highly significant difference in the DNAm-derived smoking score between smokers and non-smokers. The smoking score itself includes sites annotated to AHRR and other genes strongly associated with exposure to tobacco smoke, and so these would have been controlled for in the analysis.

[Editors' note: further revisions were suggested prior to acceptance, as described below.]

The manuscript has been improved but there are some remaining issues that need to be addressed before acceptance, as outlined below:1) The authors are asked to highlight in the Discussion what they believe is the main take-home message of the manuscript. For example, what does this meta-analysis tell us about schizophrenia that we did not know before? or about the potential use of DNA methylation in clinical study? If there are no big answers, what should be the next step forward in studying the epigenetic of schizophrenia? Because this paper is the largest meta-analysis on blood DNA methylation in schizophrenia, these sort of conclusions are expected.2) Some additional limitations should be mentioned. For example, the interaction of DNA methylation with age and sex was analyzed only for the schizophrenia-associated DMPs (which does not exclude that there could be other methylation differences associated with schizophrenia only in males or females among the remaining positions). Ignoring the possibility that genotype may interact with methylation could be another limitation of current EWASs.

We are delighted by the overall positive feedback on our revised manuscript, which was “…recognized as the largest study of DNA methylation differences in the blood of controls and patients with psychosis and considered an important piece of work…”.

We were asked to add two sections to the Discussion section before the paper could be accepted for publication. As requested, we have added a final paragraph describing more succinctly the key take-home message of the paper. Furthermore, we have extended our discussion of limitations to our study.